# Attention periodically samples competing stimuli during binocular rivalry

**Matthew J Davidson[1,2]\*, David Alais[3], Jeroen JA van Boxtel[1,2,4]†\*, Naotsugu Tsuchiya[1,2]†\***

[1]School of Psychological Sciences, Faculty of Medicine, Nursing, and Health Sciences, Monash University, Melbourne, Australia; [2]Monash Institute of Cognitive and Clinical Neurosciences, Monash University, Melbourne, Australia; [3]School of Psychology, The University of Sydney, Camperdown, Australia; [4]School of Psychology, Faculty of Health, University of Canberra, Canberra, Australia

**Abstract** The attentional sampling hypothesis suggests that attention rhythmically enhances sensory processing when attending to a single (~8 Hz), or multiple (~4 Hz) objects. Here, we investigated whether attention samples sensory representations that are not part of the conscious percept during binocular rivalry. When crossmodally cued toward a conscious image, subsequent changes in consciousness occurred at ~8 Hz, consistent with the rates of undivided attentional sampling. However, when attention was cued toward the suppressed image, changes in consciousness slowed to ~3.5 Hz, indicating the division of attention away from the conscious visual image. In the electroencephalogram, we found that at attentional sampling frequencies, the strength of inter-trial phase-coherence over fronto-temporal and parieto-occipital regions correlated with changes in perception. When cues were not task-relevant, these effects disappeared, confirming that perceptual changes were dependent upon the allocation of attention, and that attention can flexibly sample away from a conscious image in a task-dependent manner.
DOI: https://doi.org/10.7554/eLife.40868.001

**\*For correspondence:**
mjd070@gmail.com (MJD);
jeroen.van.boxtel@monash.edu (JJAB);
Naotsugu.Tsuchiya@monash.edu (NT)

†These authors contributed equally to this work

**Competing interests:** The authors declare that no competing interests exist.

## Introduction

Recent behavioral and electrophysiological evidence suggests that despite our seamless visual experience, incoming visual information is periodically enhanced for analysis in the visual system (*VanRullen, 2016a*; *VanRullen, 2016b*; *Zoefel and VanRullen, 2017*). This periodic sampling mechanism is proposed to result from the allocation of visual attention (*Busch and VanRullen, 2010*; *Dugué et al., 2016*; *Dugué and VanRullen, 2017a*; *VanRullen et al., 2007*; *Zoefel and VanRullen, 2017*), wherein alternating windows of high- and low-attentional resources operate to parcel incoming visual information, similar to the sequential frames that capture film within a video camera (*Chakravarthi and Vanrullen, 2012*; *Vanrullen and Dubois, 2011*). Whether stimuli are presented at the appropriate phase (*Busch et al., 2009*; *Mathewson et al., 2009*; *VanRullen et al., 2007*) or location (*Dugué et al., 2015*; *Dugué et al., 2016*; *Dugué and Vanrullen, 2014*; *Dugué and Van-Rullen, 2017a*; *Dugué et al., 2017b*; *Huang et al., 2015*; *Landau and Fries, 2012*; *Song et al., 2014*) of this sampling mechanism has been shown to modulate the accurate detection of a visual stimulus, in stark contrast to our experience of an uninterrupted visual environment.

To date, primary neural evidence for the rhythmic gating of visual processing stems from the dependence of target detection on the pre-target phase of neural oscillations at approximately 7–8 Hz (*Busch et al., 2009*; *Busch and VanRullen, 2010*). These spontaneous fluctuations in detection may result from the allocation of visual attention toward a single location (*Busch and VanRullen, 2010*; *Dugué et al., 2015*; *Spaak et al., 2014*; *VanRullen, 2016b*; *Zoefel and VanRullen, 2017*),

and support the assumption that neural excitability cycles gate and filter incoming information for further processing (*Schroeder and Lakatos, 2009*; *VanRullen, 2013*; *Zoefel and VanRullen, 2017*).

This periodic gating of visual perception is also prominent behaviorally in the time-course of detection accuracy. Spectral analyses applied to high temporal resolution behavioral measures reveal 7–8 Hz modulations in performance following cues to reorient attention (*Fiebelkorn et al., 2013*), which slow proportionally when attention is divided between two or more locations (e.g. *Chen et al., 2017*; *Holcombe and Chen, 2013*; *Huang et al., 2015*; *Landau and Fries, 2012*; *Landau et al., 2015*; *VanRullen, 2013*). For example, *Landau and Fries, 2012* observed that following a cue to reorient attention to either the left or right visual hemifield, target detection oscillated at a 4 Hz counterphase rhythm depending on whether cues were congruent or incongruent with the target location. Critically, this counterphase sampling of visual information persisted at ~4 Hz when attention was directed to two locations on a single object (*Fiebelkorn et al., 2013*), and when cues to reorient attention were incongruent with target location – requiring a subsequent shift in the allocation of attention to a second location (*Huang et al., 2015*). These successive fluctuations in target detection and counterphase sampling between locations have led to the suggestion that an intrinsic ~7–8 Hz attentional rhythm can be allocated over space and time in a sequential manner (*Dugué et al., 2016*; *Dugué and VanRullen, 2017a*; *Fiebelkorn et al., 2013*; *Holcombe and Chen, 2013*; *Landau and Fries, 2012*; *VanRullen, 2013*; *Zoefel and VanRullen, 2017*).

Here, we tested if rhythmic attentional sampling is at play during binocular rivalry. During binocular rivalry, incompatible images are presented to each eye which results in stochastic perceptual alternations, with one image visible at a time while the other is suppressed (*Alais, 2012*; *Alais and Blake, 2005*; *Maier et al., 2012*). In an experiment designed to induce or delay these transitions using auditory and tactile cues, we found that changes in consciousness were occurring rhythmically after the reorientation of attention. These fluctuations occurred depending on whether the crossmodal cue directed attention toward either the dominant or suppressed visual image, resulting in ~8 Hz and ~3.5 Hz oscillations, respectively. Critically, these rhythms were observed in both behavior and the electroencephalogram (EEG), and were absent when cues were not task-relevant. This approximate halving of frequency suggests that when non-visual input is inconsistent with the ongoing visual percept, attentional sampling can flexibly orient away from a consciously perceived image, seemingly 'searching for' alternative sensory information to resolve the conflict.

## Results

### Attending to low-frequency crossmodal stimulation promotes the perceptual dominance of low-frequency flicker during binocular rivalry

We manipulated the conscious visibility of images across two sessions of 24 × 3 min binocular rivalry blocks. Subjects (*N* = 34) continuously reported the content of their visual consciousness via button press to indicate which image they currently perceived, while neural activity was simultaneously recorded via 64-channel EEG (see Materials and methods). Rivalry stimuli were orthogonal sinusoidal gratings, which underwent sinusoidal contrast modulation, one at 4.5 Hz and the other at 20 Hz (*Figure 1*). In each 3 min block, we intermittently presented 12 crossmodal cues (mean duration 2.6 s), which were sinusoidally amplitude-modulated signals presented in the auditory and/or tactile modality (auditory, tactile, or combined auditory and tactile) at a frequency congruent with one of the visual stimuli (4.5 or 20 Hz). Three null cues (visual-only periods) without any crossmodal stimulation were also presented to increase the uncertainty of stimulus timing. The visual-only periods also served as a baseline to compare the behavioral effects of crossmodal cues (see below). We separated all cue periods by jittering the ISI between 7–10 s. As a result, the timing of crossmodal cues was completely independent to perceptual reports, and cues were presented at any point relative to the onset of the currently dominant percept (i.e., no closed-loop control).

In order to investigate whether the allocation of attention to crossmodal cues alters the contents of visual consciousness during binocular rivalry, we varied attentional instructions over two sessions of the experiment. For one of their two sessions (day 1 for *n* = 16, day 2 for *n* = 18), we asked subjects to count the number of times that the temporal frequency of crossmodal cues coincided with their conscious visual percept at crossmodal cue offset (see Materials and methods). For their other

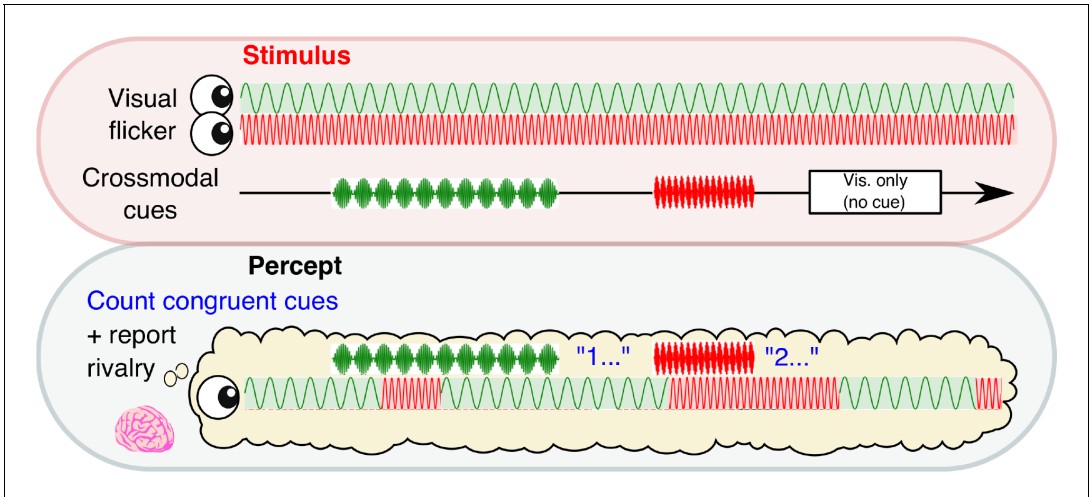

**Figure 1.** Experimental paradigm. A schematic time course showing stimulus presentation and reported visual percept. Each eye was presented with a 4.5 or 20 Hz sinusoidal flicker throughout 3 min blocks. Subjects reported their perceptual state through button-press. Crossmodal cues (also 4.5 or 20 Hz; 2, 3.1 or 4 s in duration) or visual-only periods (2.6 s in duration) were separated by inter-stimulus intervals of 7–10 s.
DOI: https://doi.org/10.7554/eLife.40868.002

session, subjects were instructed to focus on reporting their visual percept alone – ignoring any crossmodal cues.

Following the onset of a crossmodal cue, the probability of perceiving a congruent visual image increased only during attended low-frequency cues compared to all other cue types, during the period 0.68 to 3.97 s after cue onset (repeated measures ANOVAs followed by planned comparisons, FDR q = 0.05, *Figure 2a*). There was no difference in this effect when comparing the three types of crossmodal cues (auditory, tactile, and combined auditory tactile, data not shown). To confirm that this effect was due to attention, we performed a correlation-based behavioral analysis. First, we computed the Pearson's correlation coefficient (x-axis in *Figure 2b*), between each subject's verbally reported number of congruent cues (i.e., their attentional task during attend conditions), to the actual number of cues that were congruent with their visual percepts based on button-press data. Second, we defined the strength of the crossmodal cueing effect for attended low-frequency cues compared to other cue types (y-axis in *Figure 2b*), as the difference in the probability of seeing the congruent visual flicker during 1 to 4 s after cue onset. We call this the perceptual switch index (PSI), as it reflects the degree of perceptual switch after cue onset. The magnitude of these two variables displayed a strong positive correlation ($r(32) = 0.46$, $p = 0.006$, two-tailed), suggesting that the cross-modal cueing effect was indeed mediated by attention.

Due to the ongoing dynamics of binocular rivalry, this cueing effect can be calculated when visual and crossmodal information mismatched or matched at cue onset. When crossmodal cues mismatched with the visual percept at cue onset, the likelihood of switching to the previously suppressed, yet matched visual stimuli significantly increased for attended low-frequency cues compared to all other cue types over a time period from 0.62 to 4.12 s (FDR q = 0.05, *Figure 2c*). By contrast, when visual and crossmodal cues matched at cue onset, the effect of attending to low-frequency crossmodal cues delayed changes to the previously suppressed visual percept compared to all other cue types, over the period from 1.05 to 3.58 s (FDR q = 0.05, *Figure 2d*). Comparison against the visual-only cue period yielded the same conclusion, confirming that the attended low-frequency cues significantly influenced rivalry dynamics, while other cue types did not. As the overall crossmodal effects were unique to the attended low-frequency condition, we focused our subsequent attentional sampling and EEG analysis on this condition.

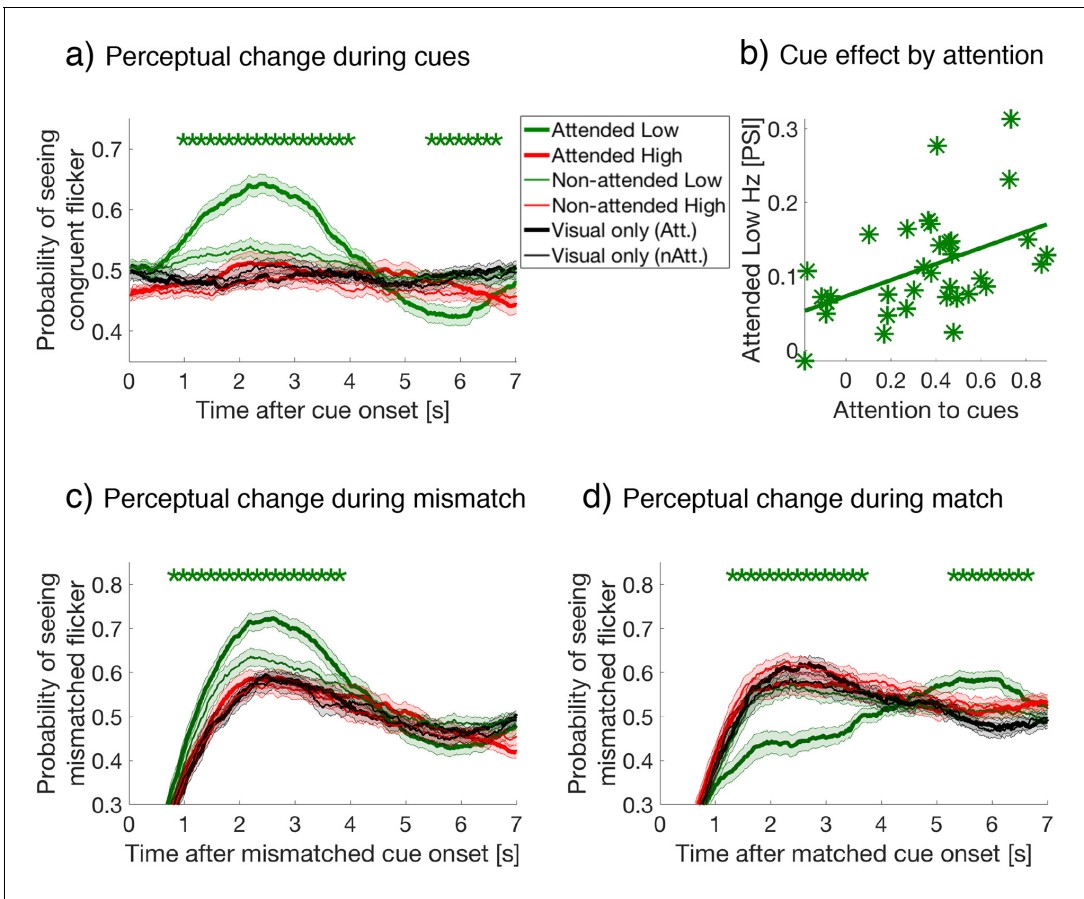

**Figure 2.** Behavioral results. (a) Button-press data, aligned at cue onset, were averaged over all crossmodal cue and visual-only periods per subject, then averaged over subjects for each cue condition. Y-axis represents the proportion of button-presses reporting congruent crossmodal and visual flicker at each time point, sampled at 60 Hz (or every 16.7 ms). Colored lines and their shading show mean ± 1 standard error across 34 subjects during attended and ignored cues (thick and thin lines) for low- and high-frequency (green and red colors). Black lines represent the equivalent probability for visual-only periods, serving as baseline (Materials and methods). Asterisks indicate a significant difference between cues at each time point (repeated-measures ANOVA followed by planned comparisons). We use FDR q = 0.05 for the statistical threshold unless noted otherwise. (b) Crossmodal effects are mediated by task-relevant attention. Our measure of crossmodal effects, the perceptual switch index (PSI, y-axis), is defined as the mean difference for the probability of seeing congruent flicker during 1–4 s after the cue onset for attended-low-frequency cues (thick green in panel a) compared to other cue types. Attention-task performance (x-axis) is the correlation coefficient between the reported and actual congruent stimuli when comparing between rivalry percepts and crossmodal cues at offset (See Materials and methods for details). The across-subject correlation between the two variables was strong ($r(32) = .46$, $p = 0.006$, two-tailed), demonstrating the crossmodal effects were strongly dependent on performance during the attention task. (c) and (d) Button-press data aligned at cue onset, with lines and shading as in panel (a). Y-axis showing the proportion of button-presses reporting the mismatched flicker at each time point, after (c) visual-crossmodal mismatch, or (d) visual-crossmodal match at cue onset. Only the data of the attended low-frequency condition differed significantly from visual-only periods.

DOI: https://doi.org/10.7554/eLife.40868.003

The following source data and figure supplements are available for figure 2:

**Source data 1.** Attended low-frequency cues alter rivalry dynamics.

DOI: https://doi.org/10.7554/eLife.40868.008

**Figure supplement 1.** Across all experimental periods, the average duration of mixed periods per switch per subject was less than 16.7 ms (our binning width), thus showing that mixed percepts are unlikely to have contributed to an increase in the variance of perceptual report timing.

DOI: https://doi.org/10.7554/eLife.40868.004

**Figure supplement 1—source data 1.** Mean mixed periods per switch per participant.

DOI: https://doi.org/10.7554/eLife.40868.005

**Figure supplement 2.** Definition of 'attention to cues' in *Figure 1c*.

DOI: https://doi.org/10.7554/eLife.40868.006

**Figure supplement 2—source data 1.** Examplary subject 'attention to cues'.

DOI: https://doi.org/10.7554/eLife.40868.007

## Binocular rivalry dynamics during attended low-frequency crossmodal cues

Our previous analysis showed that during attended low-frequency crossmodal cues, mismatched crossmodal cues lead to more perceptual switches, as the visually perceived image changed from high-frequency to low-frequency to become congruent with the crossmodal input. In the context of the attentional sampling hypothesis, we directly tested if these changes were occurring rhythmically after the reorientation of attention, and specifically investigated the timing of the first switch, defined as the first change in button-state after cue onset.

To determine if cues affected the timing of first switches, we calculated the cumulative density function of each subject's first switches after cue onset (*Figure 3a*). Compared to visual-only cue periods, first-switches after cue onset occurred earlier for mismatched cues, indicating an earlier change to the congruent, previously suppressed, visual flicker. By contrast, following matched cues first-switches during rivalry were delayed, indicating an extended maintenance of the congruent visual percept when matched with attended low-frequency crossmodal cues. The facilitation of switches by mismatched cues was observed from 0.63 to 2.45 s and 3.78 to 6.87 s relative to cue onset, with matched cues delaying switches from 1.27 to 3.77 s after onset (paired samples *t*-tests, FDR q = 0.05, in *Figure 3b*).

After cue onset, the time-course for the probability of first switches displayed rhythmic oscillatory patterns for mismatched and matched conditions (*Figure 3c and d*), but not the visual-only condition (*Figure 3e*). Each data point represents the proportion of first switches, which occurred at each time bin (16.7 ms intervals), calculated first per individual subject, and then averaged across subjects.

To quantify these patterns, we applied the Fourier transform to the period 0.5 to 2 s after cue onset (skipping the first 0.5 s to avoid an onset transient, see *Figure 3—figure supplement 1*) as performed by previous investigations of attentional sampling (*Dugué et al., 2015*; *Dugué et al., 2016*; *Fiebelkorn et al., 2013*; *Landau and Fries, 2012*). For this analyses, we corrected for multiple comparisons by using non-parametric cluster-based permutations (*Maris and Oostenveld, 2007*), with thresholds set to $p < .005$ (*Benjamin et al., 2018*) for identification within a cluster, and a final critical value for significance set to $p = 0.05$, cluster corrected (see Materials and methods).

Strikingly, when the temporal frequency of the cue matched the conscious visual flicker at cue onset, the first perceptual switches followed a 7.5–8 Hz rhythm ($p_{cluster} < 0.001$, *Figure 3f* blue), consistent with suggestions that attention samples sensory stimulation at a rate of approximately 7–8 Hz (*Dugué and VanRullen, 2017a*; *Fiebelkorn et al., 2013*; *VanRullen, 2013*). However, when crossmodal cues were mismatched with the dominant visual image at cue onset, the amplitude spectrum of perceptual switches peaked between 3.3 and 3.75 Hz ($p_{cluster} < 0.001$, *Figure 3f* magenta). This slower rhythm of perceptual changes is consistent with findings that show attention samples two locations at a rate of approximately 3.5–4 Hz (*Fiebelkorn et al., 2013*; *Landau and Fries, 2012*; *Landau et al., 2015*). No significant peaks were detected for the visual-only condition (*Figure 3f*, gray). As to the remaining three cue combinations (attended high-, ignored low- and ignored high-frequency cues), all failed to exhibit any significant crossmodal effects on perceptual switches compared to visual-only periods (shown *Figure 2a,c,d*, and *Figure 3—figure supplement 3*). Thus, we did not pursue further spectral or neural analyses of these conditions. We note that this analysis was performed on the averaged time-course, consistent with previous behavioral investigations of attentional sampling (e.g. *Fiebelkorn et al., 2013*; *Landau and Fries, 2012*). The pattern in individual participants is similar, though is not present for each individual. This is because the number of switches per condition when separating by attention/mismatch/frequency type was low, and the strength of attentional effects themselves varied across participants (*Figure 2b*).

## The neural correlates of divided and focused attentional sampling

We hypothesized that at our behaviorally observed attentional sampling frequencies (3.5 and 8 Hz), we should be able to identify the neural correlates of attentional sampling in the EEG signal using an inter-trial phase coherence (ITPC) measure. Previously, the phase of ongoing cortical oscillations have been shown to be reset by external crossmodal events (*Frey et al., 2015*; *Lakatos et al., 2009*; *Romei et al., 2012*; *van Atteveldt et al., 2014*) and to modulate the probability of target detection (*Busch et al., 2009*; *Landau et al., 2015*; *Mathewson et al., 2009*; *Thorne and Debener, 2014*; *VanRullen et al., 2007*). To isolate the specific neural correlates of attentional sampling, we

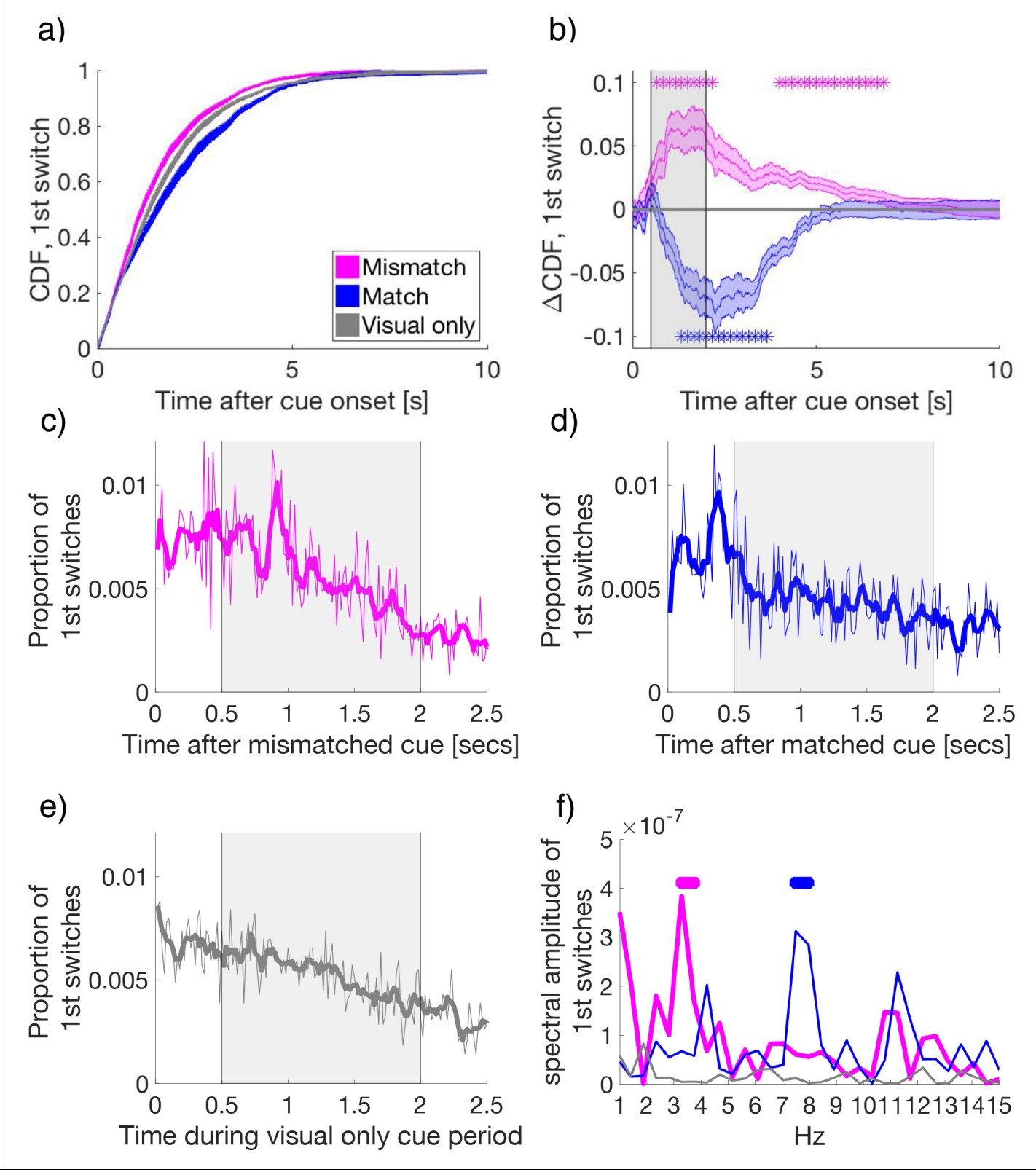

**Figure 3.** Binocular rivalry dynamics during mismatched and matched cues. (a) The cumulative density function (CDF) of the time to first-switch. Mismatched, matched, and visual-only conditions are colored in magenta, blue, and grey in all panels a–f. Lines and shading show mean and standard error across subjects (N = 34) for a and b. (b) The difference in CDFs between conditions. Asterisks mark statistical significance (paired-samples *t*-tests) comparing mismatched or matched cues to visual-only periods. FDR q = 0.05. (c–e) The time course of the proportion of first switches made after cue

*Figure 3 continued on next page*

*Figure 3 continued*

onset in (c) mismatched, (d) matched, and (e) visual-only conditions. Thin lines show the mean proportion of first-switches, binned in 16.7 ms increments and averaged across subjects. Thick lines show the smoothed data for visualization. Grey-shaded regions show the time window used for spectral analysis in (f). (f) The amplitude spectra for the time course of switches in conditions in (c-e). Asterisks indicate significant clusters (at least two neighboring frequency bins) after permutation and cluster-based corrections for multiple comparisons (see Materials and methods). The permuted null distribution and critical value for the identified clusters in f) are shown in *Figure 3—figure supplement 2*.

DOI: https://doi.org/10.7554/eLife.40868.009

The following source data and figure supplements are available for figure 3:

**Source data 1.** Source data for FIgure 3
DOI: https://doi.org/10.7554/eLife.40868.013
**Figure supplement 1.** First switches for any cues, and outside of cue periods.
DOI: https://doi.org/10.7554/eLife.40868.010
**Figure supplement 2.** The null-distributions for the surrogate datasets generated by the randomization procedure, and the actually observed values of second-stage statistics (i.e., maximum and its highest neighbor's summed Fourier amplitude).
DOI: https://doi.org/10.7554/eLife.40868.011
**Figure supplement 3.** Cumulative Density Functions for remaing crossmodal cue types
DOI: https://doi.org/10.7554/eLife.40868.012

compared the evoked ITPC, the increase in ITPC during 0 to 2 s after onset compared to −2 to 0 s before onset, in mismatched and matched cue conditions at the attentional sampling frequencies (3.5 and 8 Hz). Importantly, in these conditions, the physical sensory input was identical (i.e., attending low-frequency cues during binocular rivalry), with the only difference between conditions being the subject's percept at cue onset. Thus, any differences between conditions reflect differences due to the subjective visual percept matching or not with crossmodal cues.

For this analysis, we retained electrodes only after identification of a significant effect ($p < 0.05$, uncorrected) which also satisfied a spatial cluster-based criterion for selection and used non-parametric permutation distributions to control for multiple comparisons (*Maris and Oostenveld, 2007*; *Figure 4—figure supplement 1*). We found that the mismatched cues induced stronger ITPC than the matched cues, at 3.5 Hz over right fronto-central-temporal electrodes [FT8, C6] (*Figure 4a*) and at 8 Hz over right parietal-occipital electrodes [P6, PO8] (*Figure 5a*). *Figures 4b* and *5b* compare the evoked ITPC spectra in these regions based on mismatched and matched subjective percepts at cue onset, and confirm that our time window was long enough to distinguish the 3.5 from 4.5 Hz stimulus response (with half bandwidth = 0.5 Hz to resolve the frequency of interest).

## Attentional-sampling ITPC strength predicts perceptual outcome

Next, we investigated whether the evoked ITPC at the attentional sampling frequencies in the above-identified regions (*Figures 4a* and *5a*) predicted the magnitude of behavioral effects across subjects, shown in *Figure 2c–d*. We again computed the difference in behavioral effects when comparing attended low-frequency to all other cue types (PSI; 2 to 4 s after cue onset), as a measure for the degree of perceptual change following mismatched and matched cues. Note that when considering a wider time-window (0 to 4 s for behavioral effects, data not shown) a similar pattern of results was obtained, though weaker due to the lack of differences between cue types in early cue periods (i.e. 0 to 1 s, cf. *Figure 2c–d*). We used the evoked ITPC from 0 to 2 s after cue onset to restrict our analysis to within attended crossmodal cueing periods (which were 2, 3.1 and 4 s in duration), and to capture the period where the majority of first switches were made after cue onset (*Figure 2c and d*). Similar to the PSI, we also subtracted the evoked ITPC across all other conditions from those in the attended low-frequency condition, and abbreviate this as the normalized ITPC (nITPC) below.

In the right fronto-central-temporal electrodes ([FT8, C6]), which significantly differed in 3.5 Hz ITPC based on mismatched or matched percepts (*Figure 4a*), we found that 3.5 Hz nITPC and PSI were positively correlated for both mismatched ($r(32) = .38$, $p = 0.027$, two-tailed, *Figure 4c*), and matched cue types ($r(32) = .34$, $p = 0.049$, two-tailed, *Figure 4d*). Indicating that for both mismatched and matched cues, increases in 3.5 Hz nITPC facilitated a change in visual consciousness across subjects (*Figure 4c–d*).

In the parieto-occipital electrodes ([P6, PO8]), we found that 8 Hz nITPC was not correlated with the PSI for mismatched cues (*Figure 5a*). However, 8 Hz ITPC was negatively correlated with the PSI

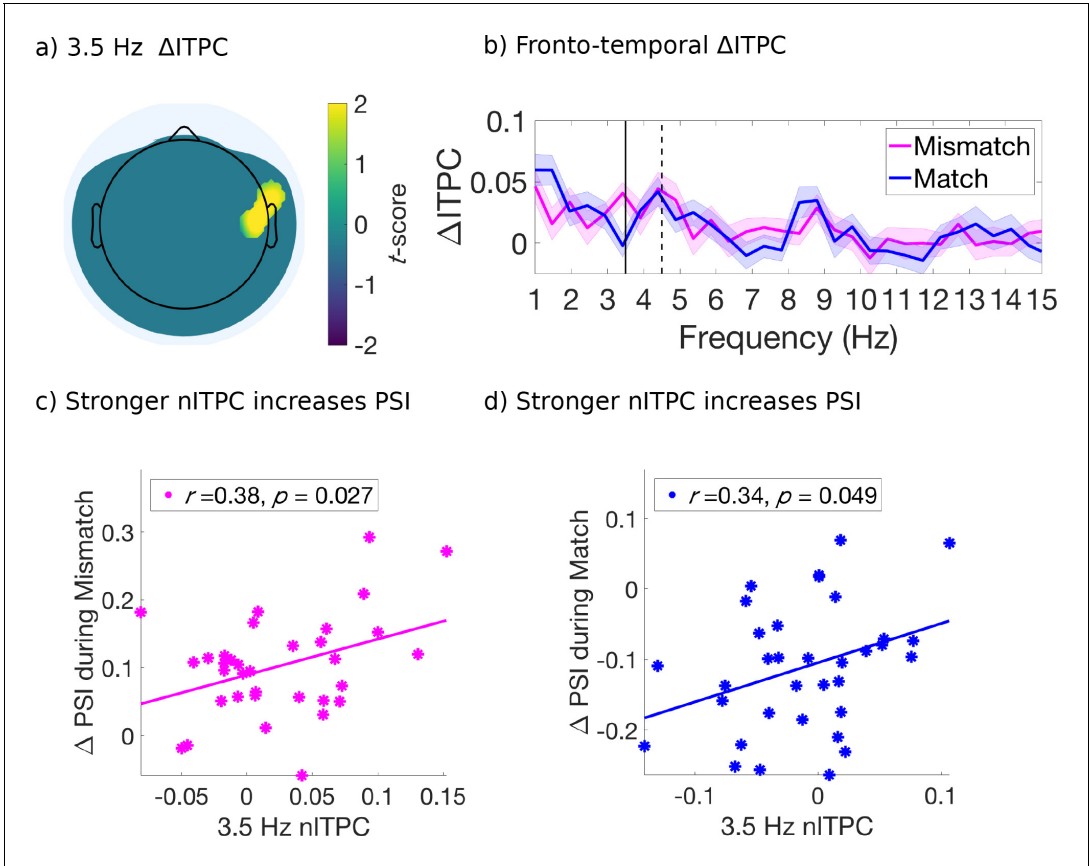

**Figure 4.** Evoked ITPC at 3.5 Hz mediates the probability of switches during mismatched and matched cues. (a) Significant differences in evoked ITPC between mismatched and matched cue conditions (multiple comparisons corrected using a cluster-based criterion; Materials and methods). Non-significant electrodes after spatial-cluster-based corrections are masked. (b) Evoked ITPC spectra at significant regions in (a). The magenta and blue lines and their shading show mean ±1 standard error of the mean across 34 subjects for mismatched and matched cues, respectively. Solid and dotted vertical black lines mark the behaviorally observed attentional sampling frequency at 3.5 Hz, stimulus frequency at 4.5 Hz respectively. (c, d): Stronger 3.5 Hz nITPC correlates with increased PSI during (c) mismatched and (d) matched conditions. The x and y-axes represent the normalized ITPC and perceptual switch index, respectively (see text for definitions). Straight lines represent least-squares regression predicting PSI from nITPC.
DOI: https://doi.org/10.7554/eLife.40868.014

The following source data and figure supplements are available for figure 4:

**Source data 1.** Source data for *Figure 4*.
DOI: https://doi.org/10.7554/eLife.40868.017
**Figure supplement 1.** Displayed are the regions selected for correlation analysis after satisfying our two-stage statistical tests on evoked ITPC, comparing mismatched and matched conditions at 3.5 Hz.
DOI: https://doi.org/10.7554/eLife.40868.015
**Figure supplement 1—source data 1.** Source data for *Figure 4—figure supplement 1*.
DOI: https://doi.org/10.7554/eLife.40868.016

during matched cues ($r(32) = -0.39$, $p = 0.023$, two-tailed, *Figure 5c*), demonstrating that increased 8 Hz nITPC resulted in fewer perceptual switches across subjects (*Figure 5d*).

## Discussion

Our findings provide novel evidence that attentional sampling exists during binocular rivalry, demonstrated in both behavior and the electroencephalogram (EEG). Behaviorally, we replicated previous evidence that stimulus-driven cues can cause a switch to previously suppressed visual stimuli when mismatched with the current percept (to bring about congruence), as well as increase the maintenance of a dominant visual image if cues matched perception (*Figure 2*; *Dieter et al., 2015*;

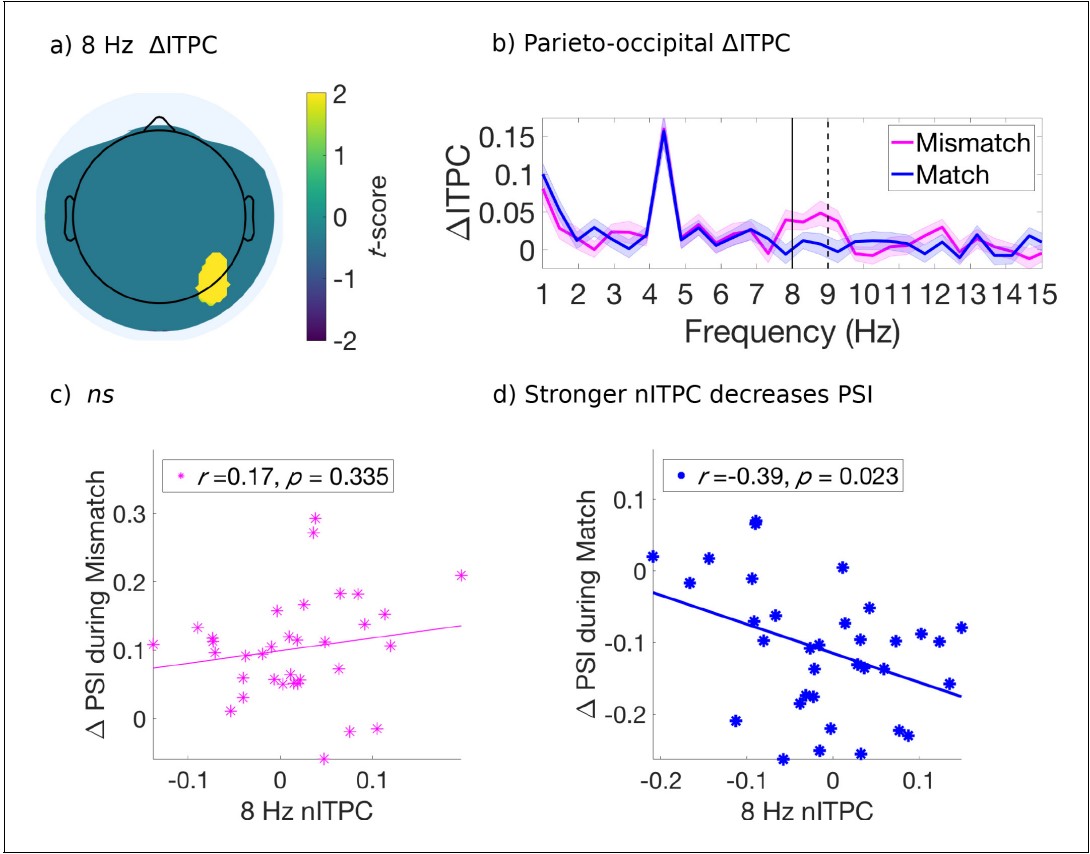

**Figure 5.** Evoked ITPC at 8 Hz mediates the probability of switches during matched cues only. (a) Significant differences in evoked ITPC between mismatched and matched cue conditions (multiple comparisons corrected using a cluster-based criterion; Methods). Non-significant electrodes after spatial-cluster-based corrections are masked. (b) Evoked ITPC spectra at significant regions in (a). The magenta and blue lines and their shading show mean ±1 standard error of the mean across 34 subjects for mismatched and matched cues, respectively. Solid and dotted vertical black lines mark in (b) the 8 Hz sampling frequency observed behaviorally and stimulus harmonic, respectively. (c,d): Stronger 8 Hz nITPC correlates with a decreased PSI for (d) matched, but not the (c) mismatched condition. The x and y-axes represent the normalized ITPC and perceptual switch index, respectively (see text for definitions). Straight lines represent least-squares regression predicting PSI from nITPC.

DOI: https://doi.org/10.7554/eLife.40868.018

The following source data and figure supplements are available for figure 5:

**Source data 1.** Source data for *Figure 5*.
DOI: https://doi.org/10.7554/eLife.40868.021
**Figure supplement 1.** Displayed are the regions selected for correlation analysis after satisfying our two-stage statistical tests on evoked ITPC, comparing mismatched and matched conditions at 8 Hz.
DOI: https://doi.org/10.7554/eLife.40868.019
**Figure supplement 1—source data 1.** Source data for *Figure 5—figure supplement 1*.
DOI: https://doi.org/10.7554/eLife.40868.020

*Lunghi and Alais, 2015*; *Lunghi et al., 2014*). Critically, we found distinct attentional sampling frequencies evident in the time-course of first switches during these cues (*Figure 3*). When crossmodal cues were incongruent in temporal frequency with the dominant visual stimulus, switches in visual consciousness occurred earlier, and within a distinct ~3.5 Hz rhythm. This 3.5 Hz rhythm is consistent with previous reports of divided attentional sampling between two locations (*Fiebelkorn et al., 2013*; *Landau and Fries, 2012*; *Landau et al., 2015*). However, when crossmodal cues were matched in temporal frequency to the dominant visual stimulus, changes in visual consciousness demonstrated an ~8 Hz rhythm, consistent with periodicities in behavioral measures observed when attending to a single visual location (*Fiebelkorn et al., 2013*), and suggestions that a cortical 7–8 Hz attentional rhythm may gate visual processing (*Busch and VanRullen, 2010*; *Dugué and VanRullen, 2017a*; *Fries, 2015*). In the EEG (*Figures 4* and *5*), distinct correlates of these divided and focused

attentional sampling frequencies emerged over right fronto-temporal and right parieto-occipital sites, respectively, with ITPC strength at these frequencies correlating with the behaviorally reported change in consciousness across subjects.

Traditionally, top-down, voluntary attention has been thought to have limited control over perceptual dynamics during binocular rivalry; attention may alter dominance durations, but cannot halt the process of perceptual reversals entirely (*Chong and Blake, 2006*; *Chong et al., 2005*; *Chopin and Mamassian, 2010*; *Dieter et al., 2016b*; *Dieter et al., 2015*; *Dieter et al., 2016b*; *Mitchell et al., 2004*; *Paffen and Alais, 2011*; for bottom-up control, including crossmodal stimulation, see *Conrad et al., 2010*; *Deroy et al., 2014*; *Guzman-Martinez et al., 2012*; *Kang and Blake, 2005*; *Lunghi and Alais, 2013*; *Lunghi et al., 2010*; *Lunghi et al., 2014*; *van Ee et al., 2009*). Our results clearly show additional dependence on the top-down deployment of attention, as without explicit instruction to attend to crossmodal signals, no facilitatory crossmodal effects emerged (see also *Jack and Hacker, 2014*; *Talsma et al., 2010*; *van Ee et al., 2009*). This interaction between low-level stimulus features (temporal frequency) and the allocation of attention indicates the facilitative role of both crossmodal stimuli (*Deroy et al., 2014*; *Deroy et al., 2016*) and attention for perceptual transitions during binocular rivalry (*Dieter et al., 2016a*; *Dieter et al., 2015*; *Dieter and Tadin, 2011*; *Paffen and Alais, 2011*; *Zhang et al., 2011b*). Our results are consistent with previous research, which has shown that exogenous feature-based cues can bias rivalry dynamics (*Dieter et al., 2015*; *van Ee et al., 2009*) and extend these reports by revealing an oscillatory basis to these changes in visual perception.

Our demonstration of these oscillatory changes in visual consciousness, which have been evoked by attended crossmodal cues, are also relevant to current computational models of binocular rivalry (e.g. *Laing and Chow, 2002*; *Li et al., 2017*; *Wilson, 2003*). No current model has accounted for the interaction we observe between attentional allocation and crossmodal stimuli, nor attention as an oscillatory process. Most recently, *Li et al., 2017* have described a model for binocular rivalry incorporating attention and mutual inhibition. In their model, attentional modulation is dealt to the sensory representation that has the stronger sensory responses, by providing feedback to monocular-excitatory drives that otherwise increase monotonically with stimulus contrast. Building on our findings, future models could incorporate an oscillatory increase in excitatory drive as a result of periodic, rather than sustained attentional modulation (*Fiebelkorn et al., 2018*; *Helfrich et al., 2018*; *VanRullen, 2018*).

Previous behavioral investigations of attentional sampling have relied upon a brief cue to reorient attention, before estimating the time-course of target detection by densely sampling subject responses over closely spaced target-presentation intervals. Our design is unique in that 'target-detection' here is operationalized as the first reported change in visual consciousness for a continuously presented stimulus, resolved at 16.7 ms (or 60 Hz) from 500 ms to 2000 ms following cue-onset.

Past researchers have demonstrated approximately 7–8 Hz fluctuations in perceptual performance following the allocation of visual attention (*Dugué et al., 2015*; *Fiebelkorn et al., 2013*; *VanRullen, 2013*; *VanRullen et al., 2007*; *Zoefel and VanRullen, 2017*), commensurate with suggestions that cortical oscillations at approximately 7–8 Hz gate the content of visual perception (*Busch and VanRullen, 2010*; *Dugué and VanRullen, 2017a*; *Hanslmayr et al., 2013*). In our binocular rivalry paradigm, we also observed changes in visual consciousness occurring within an 8 Hz rhythm, yet unique to when cues were congruent with the dominant visual percept at cue onset. By contrast, perceptual sampling has previously been observed at ~4 Hz when cues have encouraged dividing attention between two objects or locations (*Dugué et al., 2016*; *Fiebelkorn et al., 2013*; *Huang et al., 2015*; *Landau and Fries, 2012*; *Landau et al., 2015*; *Song et al., 2014*). As such, the ~3.5 Hz rhythm we observed when crossmodal cues mismatched with the conscious visual percept extends the evidence for divided attentional sampling to binocular rivalry.

From our data alone, we cannot infer whether during conventional binocular rivalry, attention samples at 8 or 4 Hz. We surmise that increased attention to stimulus competition may be required to observe the attentional sampling rhythms we report here. Indeed outside of attended cue periods we did not observe periodic behavioral responses (*Figure 3—figure supplement 1*), suggesting that attentional sampling is commensurate with sustained and goal-directed attention, instead of persisting throughout rivalry. We also note that the issue of trial-to-trial variability when reporting on perceptual changes cannot be completely avoided in binocular rivalry research, and is important to

consider. Here, one might argue that variable timing in perceptual reports may blur any effects of temporal periodicity. However, our results clearly demonstrate that a change in perceptual state occurred for attended low-frequency cues that were unique in changing visual consciousness compared to visual-only baseline, at or near frequencies of attentional sampling that have been reported in previous literature. By comparison, the time-course of visual switches during visual-only periods did not exhibit periodic sampling, nor did the first-switches during non-attended, or high-frequency cues.

Distinct neural correlates of these attentional sampling rhythms were also found in the EEG. We found significantly greater 3.5 Hz ITPC strength for mismatched compared to matched cue types over right fronto-centro-temporal electrodes [FT8 and C6], suggesting this region may be a candidate neural correlate for divided periodic attentional sampling (*Figure 4a*). Accordingly, following both mismatched and matched cues, increased 3.5 Hz ITPC in this region also positively correlated with the likelihood of switching to the previously suppressed visual image across subjects (*Figure 4c–d*). Using visual-only stimulation, previous research has identified a pre-target ~4 Hz phase-dependency for peri-threshold perception when attention is divided across visual hemifields (*Landau et al., 2015*). While our right fronto-temporal region is different to those previously implicated in attentional-sampling (e.g. *Landau et al., 2015*), we note that in our paradigm, attention was not divided between visual hemifields, but between competing stimuli during binocular rivalry. Right fronto-temporal regions have previously been implicated in the reorientation of attention to unattended locations (*Corbetta and Shulman, 2002*; *Downar et al., 2000*; *Proskovec et al., 2018*), and most recently, *Helfrich et al., 2018* have demonstrated that the phase of ~4 Hz dependent sampling in frontal and parietal areas determines visual perception. Taken together, our results support previous research indicating that a periodic attentional sampling mechanism modulates visual perception, here extending this finding into binocular rivalry when visual stimuli spatially overlap and compete for perceptual dominance.

We also found behavioral and neural correlates of focused attentional sampling during binocular rivalry when cues were consistent with the prevailing visual percept. Specifically, 8 Hz ITPC over parieto-occipital electrodes was negatively correlated with the likelihood of switching to the incongruent perceptual outcome (*Figure 5d*). Previously, phase-dependent peri-threshold perception has been reported for focused attention tasks in the visual domain (*Busch et al., 2009*; *Busch and VanRullen, 2010*; *Hanslmayr et al., 2013*; *Mathewson et al., 2009*), and has primarily implicated an approximately 7 Hz component located over fronto-central electrodes (*Busch et al., 2009*; *Busch and VanRullen, 2010*). Given the differences between paradigms, it is unsurprising that our identified region for focused attentional sampling does not coincide with those reported in previous research regarding phase-dependent perception. Particularly as our right-lateralized response may be due to the left-lateralized tactile input used to investigate crossmodal attentional sampling (though ITPC was not different among the three crossmodal stimulation types, data not shown). While promising, future experiments that control for this lateralization are needed to characterize the contributions of fronto-centro-temporal and parieto-occipital regions to this effect, particularly as activity over each of these regions has previously been implicated in the reorienting of visuo-spatial attention (*Corbetta and Shulman, 2002*; *Downar et al., 2000*; *Dugué et al., 2018*; *Proskovec et al., 2018*), and for the integration of multisensory stimuli into a coherent percept (*Beauchamp, 2005*; *Bushara et al., 2003*; *Calvert and Thesen, 2004*; *Driver and Noesselt, 2008*; *Zhang et al., 2011a*). Increases in right parieto-occipital theta power (4–8 Hz) have also been shown when attending to visual stimuli in the presence of auditory distractors (*van Driel et al., 2014*), with the phase of right parieto-occipital alpha (8–10 Hz) or theta (6–7 Hz) oscillations determining the perceptual outcome of bistable stimuli (*Ronconi et al., 2017*). As such, the present modulation for 8 Hz parieto-occipital ITPC is consistent with the idea that right-parietal networks may preferentially represent temporal information in the visual modality (*Battelli et al., 2007*; *Guggisberg et al., 2011*).

Reporting binocular rivalry switches involves both a change in perception, and a decision to press the button. Accordingly, the attentional sampling in binocular rivalry we report here may reflect the fluctuation of perception or of decision criterion. Recent studies of behavioral oscillations that have employed signal detection theory have reported that sensitivity and response criterion both exhibit oscillations (at distinct frequencies) in the high theta/low alpha band, for both vision (*Zhang et al., 2018*) and audition (*Ho et al., 2017*). Consequently, whether our oscillations reflect perceptual or decision-level effects must be clarified (*Ho et al., 2017*; *Iemi and Busch, 2018*; *Iemi et al., 2017*;

*Limbach and Corballis, 2016*). Our paradigm cannot resolve this as it is, although a future investigation using our paradigm combined with signal detection theory could do so.

Our analysis has so far revealed that when crossmodal cues mismatched with the dominant binocular rivalry stimulus, that rates of attentional sampling slowed to ~3.5 Hz – implicating the division of attention over multiple locations. However, our exogenous cues oriented attention toward the congruency of visual and crossmodal stimuli, prompting the question: between what was attentional sampling divided? One possibility is that attentional sampling during mismatched cues was divided between two sensory modalities, as the brain tried to resolve a conflict between concurrent auditory/tactile and visual information. *Figure 6a* provides a schematic of this multisensory interpretation. If the neural activity in our identified region is representative of divided sampling between modalities, it constitutes the first evidence that an attentional sampling mechanism can flexibly orient between temporally co-modulating crossmodal stimuli. Although the facilitative role of attention in multisensory integration remains controversial (*Hartcher-O'Brien et al., 2016*; *Talsma et al., 2010*), we see it as a viable possibility that this mechanism resolved perceptual ambiguity through a visual perceptual switch to the competing image, rendering the multisensory stimuli congruent.

As only attended, low-frequency modulated cues enabled a change in visual consciousness, we must consider whether the lack of a high-frequency effect reflects an upper limit in temporal frequency on crossmodal interactions or attention. Such a limit on crossmodal interactions may explain why we observed low- but not high-frequency behavioral effects in the present task, and is supported by previous investigations regarding the binding of multisensory stimulus attributes (*Fujisaki and Nishida, 2005*; *Fujisaki and Nishida, 2010*; *Lunghi et al., 2014*; *Vroomen and Keetels, 2010*), and the limits of crossmodal temporal judgments (*Fujisaki and Nishida, 2005*; *Fujisaki and Nishida, 2010*; *Holcombe, 2009*; *Vroomen and Keetels, 2010*). For example, *Fujisaki and Nishida, 2005* have shown that judgments of temporal synchrony between rhythmic sensory streams degrade above ~4 Hz. It is plausible that the ineffective crossmodal cueing that we found is related to the above-mentioned findings, reflecting a limit on crossmodal integration processes, rather than attention.

Having said that, one previous study using a similar design to ours was successful in eliciting a high-frequency crossmodal effect (15–20 Hz; *Lunghi et al., 2014*), and notably, in the absence of explicit attentional demands. However, these differences are not wholly unexpected, as to optimize the present task for EEG recordings we used larger (6.5° visual angle) luminance-modulated sinusoidal gratings to facilitate subsequent steady-state visually evoked potential analyses. While in comparison, Lunghi et al. succeeded in showing a high-frequency effect with rivalry stimuli that were contrast-modulated narrow-band random noise patterns (3.2° visual angle), and did so under conditions analogous to our non-attend conditions. This difference in the composition of visual stimuli is noteworthy, as stimulus size is known to strongly affect rivalry dynamics (*Blake et al., 1992*). To our knowledge, whether stimulus size impacts upon crossmodal effects during binocular rivalry is unknown. However, given the strength of our results for attended low-frequency flicker (*Figure 2a*), we note that the low- and high-frequency effects observed by *Lunghi et al., 2014* are not generalizable to the larger and attended-rivalry stimuli employed here. Similarly, whether the type of stimuli (e.g., gratings vs random noise patterns) also impacts upon crossmodal effects during rivalry represents a fruitful endeavor for research, particularly given the novel possibility of distinguishing between crossmodal and attentional limits on attentional sampling.

An alternate possibility to crossmodal attentional sampling is that the 3.5 Hz rhythm in our paradigm reflects divided attentional sampling between dominant and suppressed visual images during binocular rivalry (*Figure 6b*). The frequency of divided attentional sampling that we observed is consistent with those obtained when visual attention has been divided between two objects or locations (*Fiebelkorn et al., 2013*; *Landau and Fries, 2012*). As our binocular rivalry stimuli necessarily occupied the same spatial location, attention in our paradigm was likely divided between either features or objects, instead of locations. Indeed, feature-based attention has already been shown to modulate neural processes when an attended target is suppressed during continuous flash suppression (*Kanai et al., 2006*). During binocular rivalry, perceptual dominance is also influenced by object-based attention (*Mitchell et al., 2004*), with unconscious selection mechanisms argued to mediate perceptual transitions (*Lin and He, 2009*). This second alternative is also indirectly supported by the temporal limits of binocular rivalry when conflicting visual stimuli are presented asynchronously, without temporal overlap between the two eyes (*O'Shea and Blake, 1986*; *van Boxtel et al., 2008b*;

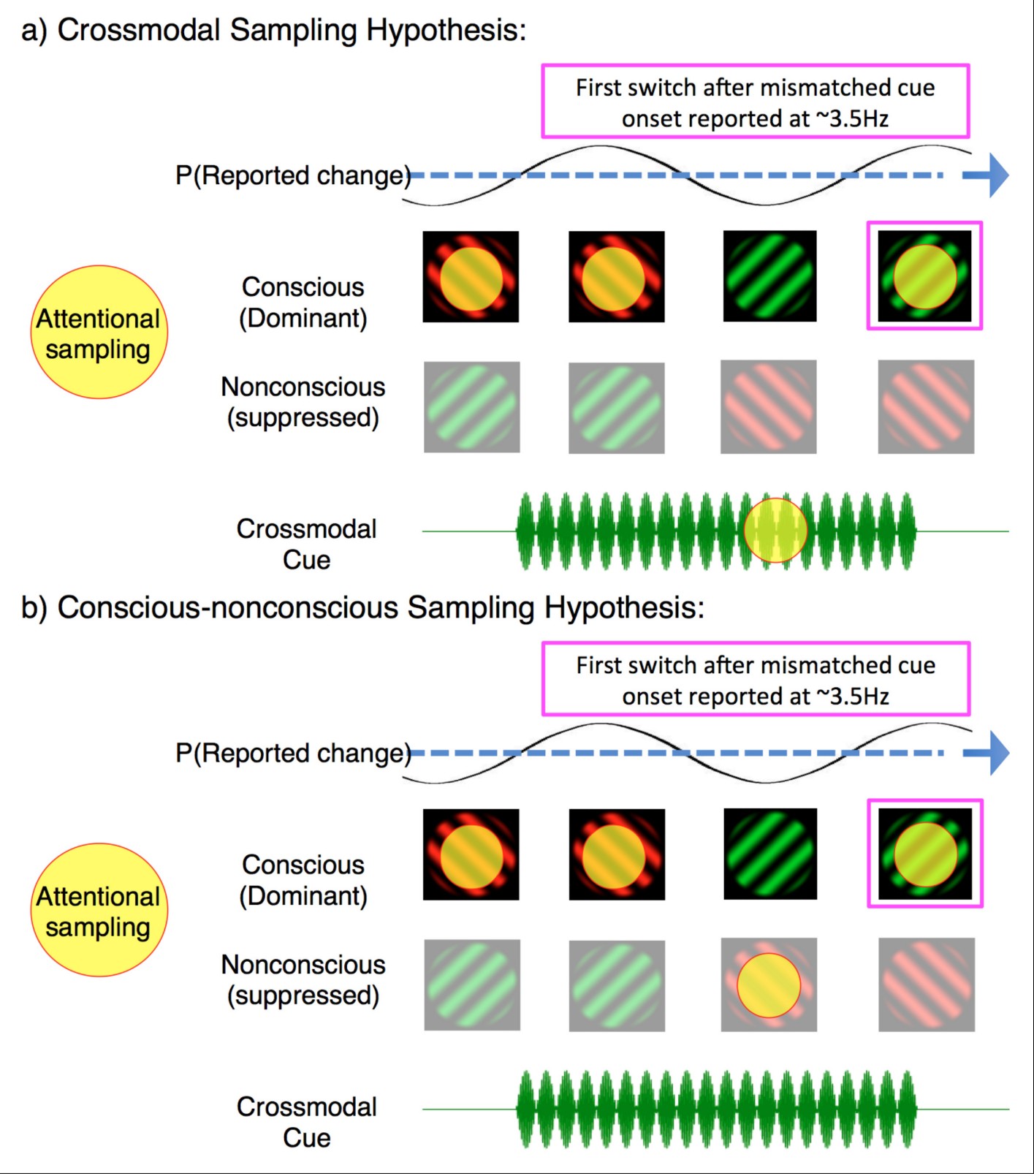

**Figure 6.** Two possible interpretations of attentional sampling during mismatched crossmodal cues. Schematic representation of attentional sampling and perceptual oscillations during binocular rivalry. (a) Crossmodal sampling hypothesis: While perceiving the high-frequency visual flicker, an attended low-frequency crossmodal cue mobilises attention to sample between the dominant image and mismatched crossmodal cue at ~3.5 Hz. As a consequence, the likelihood of the first perceptual switch is modulated at ~3.5 Hz. (b) Conscious-nonconscious sampling hypothesis: The onset of a

*Figure 6 continued on next page*

*Figure 6 continued*

mismatched cue prompts attention to sample between separate visual features, which in our paradigm consists of dominant and suppressed visual images. We do not suggest that these are the only mechanisms of attentional sampling during binocular rivalry, and only illustrate the interpretations discussed.

DOI: https://doi.org/10.7554/eLife.40868.022

*van Boxtel et al., 2007*). The maximum stimulus onset asynchrony that can sustain this type of rivalry is approximately 350 $\pm$ 50 ms, beyond which alternating stimuli introduced to one eye are perceived immediately, without rivalry occurring (*van Boxtel et al., 2008a*). This limit is consistent with a 7–8 Hz attentional sampling rhythm distributed between the two conflicting stimuli (each sampled at ~3–4 Hz). When stimuli are presented rapidly enough they are temporally bound together and can engage in ongoing rivalry; when stimuli are presented slower than at 3–4 Hz, they are temporally individuated by attention, and rivalry ceases. A recent computational model that explicitly modeled time-varying attention could indeed reproduce this finding (*Li et al., 2017*), suggesting that attention is the process that temporally binds the successive stimulus presentations together. Based on our findings, we propose the persistence of percepts may also be modeled using an oscillatory, and feature-selective attentional mechanism (*Li et al., 2015*; *Li et al., 2017*).

The suggestion that attention can sample between conscious and nonconscious vision is also consistent with a view that the underlying neuronal processes for attention and consciousness are supported by distinct neural mechanisms (*Bahrami et al., 2007*; *Watanabe et al., 2011*); for review see *Koch and Tsuchiya, 2007*). We note that while attentional sampling of a suppressed image suggests that attention is not sufficient for consciousness (*Dehaene et al., 2006*; *Koch and Tsuchiya, 2007*; *Lamme, 2003*; *van Boxtel et al., 2010a*; *van Boxtel et al., 2010b*), this interpretation remains consistent with a view that attention may still be necessary for conscious perception (*Chica and Bartolomeo, 2012*; *Cohen and Dennett, 2011*; *Merikle and Joordens, 1997*; *O'Regan and Noë, 2001*; *Posner, 1994*; *Posner, 2012*).

Whether attributable to conscious-nonconscious, or visual-crossmodal attentional sampling, the present results also complement the 'active-sensing' hypothesis (*Schroeder et al., 2010*), whereby perceptual selection is determined by routine exploratory behaviors. According to the active-sensing hypothesis, attention is critical to search for task-relevant information from the environment (*Schroeder et al., 2010*), particularly via the rhythmic coordination of multisensory information (*Schroeder et al., 2010*; *Thorne and Debener, 2014*). Intriguingly, early contributions from multisensory (non-visual) information have been shown to modulate perception (*Morillon et al., 2014*; *Schroeder et al., 2010*; *van Atteveldt et al., 2014*). The rhythmic modulation of visual performance has also been demonstrated to follow the onset of both voluntary (*Hogendoorn, 2016*), and preparatory motor behaviors (*Tomassini et al., 2017*; *Tomassini et al., 2015*). Here, in further support of the active-sensing hypothesis, we have shown that task-relevant crossmodal information can change the rhythmic modulations of perceptual selection during competition for perceptual dominance.

In summary, here we have provided novel evidence in support of attentional sampling during binocular rivalry through the use of crossmodal cues matched to either a conscious or nonconscious visual stimulus. As the attention sampling hypothesis continues to garner traction from various psychophysical and neuronal paradigms (*Fiebelkorn et al., 2018*; *Helfrich et al., 2018*; *VanRullen, 2016a*; *VanRullen, 2016b*; *VanRullen, 2018*), future targeted experimentation can confirm whether attention can indeed sample across modalities (*Figure 6a*), as well as if attention can sample between conscious and nonconscious neural representations during binocular rivalry (*Figure 6b*). The interactions between crossmodal stimuli and conscious perception represent a fruitful avenue for experimentation (*Faivre et al., 2017*), here uncovering the previously unknown dependence of attention and consciousness on rhythmic neural dynamics of the human brain.

## Materials and methods

### Participants

A total of 34 healthy individuals (21 females, 1 left-handed, average age 23 $\pm$ 4.7) were recruited via convenience sampling at Monash University, Melbourne, Australia. All had normal or corrected-to-

normal vision and gave written informed consent prior to participation. Monash University Human Research and Ethics Committee approved this study, and subjects were paid 15 AUD per hour of their time, over an approximate total of 5 hours.

## Apparatus and stimuli

Stimuli were generated using Psychtoolbox (*Brainard, 1997*); RRID:SCR_002881) and custom MAT-LAB scripts (https://github.com/Davidson-MJ/BRproject-attentionsampling; RRID:SCR_ 001622; *Davidson, 2018*). Each visual stimulus was viewed through a mirror stereoscope placed at an approximate viewing distance of 55 cm from computer screen (29 × 51 cm, 1080 × 1920 pixels, 60 Hz refresh rate) with the subject's head stabilized via chin rest. Rivalry stimuli were red and green gratings displayed on a black background, with a white frame to aid binocular fusion, embedded within the wider gray background of the remaining portions of the screen. Beside each white framed image, colored arrows indicated the direction for button-press (e.g., right for red, left for green). Gratings were sinusoidal with spatial frequency of 0.62 cycles per degree, oriented ±45° from vertical, and subtended 6.5° visual angle (240 × 240 pixels on the display). Visual stimuli were sinusoidally contrast-modulated at either 4.5 or 20 Hz using a temporal sinusoidal envelope. The phase of each grating was static throughout each 3 min binocular rivalry block, yet shifted after each block to reduce the effects of visual adaptation. The stimulus size was chosen after piloting the largest images that could support minimal incidences of piecemeal rivalry. The very low spatial frequency of 0.6 cycles per degree and the rapid temporal modulations both favor neurons with large receptive fields and thus reduce the incidence of piecemeal rivalry. In addition, by rivaling red and green stimuli, each image had a consistent color which helps group rivalry alternations and maintain perceptual coherence rather than piecemeal switching. We explained to participants that piecemeal percepts may occur and in such cases they should indicate the stimulus that was most predominant (see *Figure 2—figure supplement 1*).

For crossmodal stimuli 50 Hz carrier tones were amplitude modulated by 4.5 or 20 Hz sine waves to create digital waveforms, which were either 2, 3.1 or 4 s in duration. For tactile stimulation, subjects clasped a wooden ball with their left hand attached to a Clark Synthesis Tactile Sound Transducer (TST429 platinum) housed in a custom sound insulated box (*Lunghi et al., 2014*). Auditory stimulation was delivered binaurally through Etymotic HD5 noise reduction headphones, with ACCU-Fit foam ear tips to reduce ambient noise.

## Stimulus timing

Accurate stimulus timing of synchronous visual and crossmodal stimuli was ensured with a WDM-compatible, hardware-independent, low-latency ASIO driver (www.asio4all.com), which was necessary to minimize audio buffer duration to sub-millisecond intervals and reduce latency compensation. The time-course of stimulus presentation was also physically recorded in the EEG for offline analysis. Photodiodes were used to record the flicker-envelope of visual stimuli and stored as separate channels in the ongoing EEG. The waveforms for crossmodal stimulation were simultaneously sent to both the presentation hardware and external electrode channels using a digital splitter (Redback A2630 4 Channel Headphone Distribution Amplifier). Stimulus presentation lag was assessed by computing the difference between the recorded frames of trigger-codes and actual physical trace within the EEG as part of data pre-processing. We adjusted the relative timing of behavioral and EEG data accordingly as part of this analysis. In most cases, no adjustment was necessary, requiring a maximum change of 3 frames in duration on <1% of blocks across all subjects.

## Calibration of visual stimuli

A maximum of 10 one-minute binocular rivalry blocks were performed prior to experimentation on the first day for all subjects. These blocks served to familiarize subjects with reporting their visual percepts during binocular rivalry, and to calibrate approximately equal dominance durations for the flickering stimuli in each eye. Contrast values for left/right eye, green/red color, and low/high frequency stimulus combinations (in total, eight combinations) were adjusted on a logarithmic scale until approximately equivalent total dominance durations were reached (between 1:1 and 1:1.5), with the additional requirement that the average perceptual duration for each stimulus was longer than 1 s. As there were 24 unique 3 min binocular rivalry blocks on each day of experimentation,

each of the 8 combinations of visual parameters were balanced across all three crossmodal conditions.

## Calibration of auditory stimuli

Prior to experimentation, subjects were also tasked with equating the perceptual intensity of tactile and auditory stimulation for each low- and high-frequency condition, to achieve approximately equal phenomenological intensity across subjects and stimulus conditions. For all subjects, the amplitude of tactile vibrations was set to the same comfortable, supra-threshold level (approximately equivalent to 65 dB SPL). In the absence of visual stimulation, simultaneous auditory and tactile stimuli were then presented in a staircase procedure, with subjects adjusting the amplitude of auditory tones to match the perceived intensity of simultaneous tactile vibrations. They performed the matching task separately within low-frequency auditory tones and tactile vibrations and within high-frequency auditory tones and tactile vibrations. This calibration procedure was performed on each day of testing, to account for differences in the insertion depth of inner-ear headphones across separate days.

## Experimental procedure and behavioral analysis

A total of 24 three-minute binocular rivalry blocks were presented on each of the two separate days of testing. In each block, subjects reported their dominant visual percept during rivalry while receiving occasional crossmodal cues, which were either auditory, tactile, or simultaneous auditory and tactile. In a given three-minute block, we presented only one of the three types of crossmodal cues. The order of these blocks were randomized for each subject and each day of experimentation. In each block, 12 trials of crossmodal cues were presented. Each cue was either low (4.5 Hz) or high (20 Hz) frequency auditory and/or tactile stimulation. Six cues were presented for each frequency, with durations composed of three x 2 s, two x 3.1 s, and one x 4 s cues. To increase uncertainty for the timing of the cues, we inserted three null cues (which we call visual-only periods, *Figure 1*) without any crossmodal stimulation for a duration of 2.6 s (the average duration of crossmodal cues). We also used these visual-only periods as a baseline for behavioral analyses (*Figures 2* and *3*). We randomized the order of all cues, which were separated with uniform jittering by 7–10 s ISI within each block.

Across all sessions, subjects were told to focus on accurately reporting their dominant visual percept at all times via button-press. As the state of the button-press was sampled at 60 Hz, the same rate as the video refresh rate, we were able to estimate the probability and time-course of binocular rivalry dynamics over 16.7 ms intervals.

Over two sessions on separate days, subjects distributed attention between visual rivalry and crossmodal cues based on separate task instructions. On day 1 for *n* = 18 or day 2 for *n* = 16, subjects were instructed to ignore the crossmodal cues and to focus on reporting only visual rivalry. For the other session, subjects were instructed to distribute attention across both visual rivalry and crossmodal cues. To ensure their attention was on task, these alternate days included task instructions for subjects to silently tally the number of times the temporal frequency of an attended crossmodal cue matched that of their dominant visual percept at the time of crossmodal cue's offset. Due to the varied duration of crossmodal cues, this task ensured that attention was allocated consistently throughout the presentation of crossmodal cues. To familiarize subjects with these task demands, an additional two practice blocks (3 minutes each) were included during the calibration procedure on the relevant day of experimentation. Although 34 subjects were retained for final analysis, five others were recruited and began the experiment, yet failed to complete their second day of experimentation. One other subject was removed due to their failure in following task instructions and excessive movement during EEG recording.

## Evaluation of attention-on-task

To evaluate the attentional allocation to both visual and crossmodal stimuli, at the end of each 3 min block we asked subjects to verbally report their subjective estimate of the number of crossmodal stimuli which were matched in temporal frequency to the flicker of their dominant visual percept at the point of attended crossmodal cue offset. Then, we defined an index, 'attention to cues' (*Figure 2b,x*-axis) as the correlation coefficient between 24 subjective estimates (one per attended

block) and the actual recorded occurrences of congruent stimuli. *Figure 2—figure supplement 2* displays the correlation between subjective and actual congruent stimuli for an exemplary subject.

## Behavioral data analysis

We preprocessed the button-press data to accurately estimate the timing of changes in visual consciousness during binocular rivalry. First, we categorised each time-point according to the flicker frequency of the dominant visual stimulus reported. To analyze the time-course of the probability of a button-press state (*Figure 2a*), we categorized button-presses (which could correspond to either low- or high-frequency) as either congruent or incongruent with the ongoing crossmodal stimulus frequency. Then, we obtained the probability of a congruent button-press state as a function of time per subject, by averaging responses at each time point across all 144 trials per attention x frequency cue subtype.

For visual-only periods, the left button (corresponding to left-eye dominance) was arbitrarily set to congruent prior to the averaging of probability traces within subjects. As visual parameters were balanced across all blocks, this selection necessarily balanced across visual frequency and color parameters, and we note that the identical analysis performed using right-eye congruence produced equivalent results. Mismatched (*Figure 2c*) or matched (*Figure 2d*) condition comparisons were then defined by whether the congruent button (left-eye dominant) was pressed at cue onset.

In *Figure 2c and d*, we set the y-axis for 'Probability of seeing mismatched flicker', to reflect the probability of perceptual states that differ in temporal frequency from the crossmodal cue. In *Figure 2a,c and d*, we compared among six conditions with one-way repeated-measures ANOVAs: 1, visual-only on attend days; 2, visual-only on non-attend days; 3, attended low-frequency; 4, attended high-frequency; 5, unattended low-frequency; and 6, unattended high-frequency. We defined significant differences among conditions at those time points that survived corrections for multiple comparisons with planned comparisons between cue types and the visual-only baseline, using FDR at q = 0.05 (*Benjamini and Yekutieli, 2001*).

## Perceptual switch index (PSI)

To quantify crossmodal effects during binocular rivalry, we defined the perceptual switch index (PSI). PSI is the difference in the probability of a change in percept when comparing attended low-frequency to four other crossmodal cues. For the y-axis in *Figure 2b*, we calculated the PSI as the difference in the probability of viewing a congruent visual flicker over the period 1–4 s after stimulus onset. The same subtraction was used to compare the probability of viewing the previously suppressed visual flicker following mismatched (*Figures 4c* and *5c*) and matched cues (*Figures 4d* and *5d*), for the period 2–4 s after onset. This shorter time window was selected to capture when the crossmodal effects on binocular rivalry emerged for mismatched and matched cues. A similar pattern to the results displayed in *Figures 4* and *5* was shown when a wider window was used (e.g. 0–4 s, data not shown).

## Spectral analysis of first switches

For our spectral analysis (*Figure 3*), we focused on the first perceptual switches, which were the first time-point recording a change in button-press state after cue onset. To account for individual variation in the amount of overall switches, the proportion of switches at each time point was first calculated at the subject level, before averaging across all subjects. We sampled button-presses at 60 Hz (or every 16.7 ms). For the spectral analysis of perceptual switches (*Figure 3f*), we applied a single-taper fast Fourier transform (FFT) to the period 0.5–2 s after cue onset (Nyquist = 30 Hz, a half bandwidth = 0.67 Hz). This window was selected to restrict the analysis so that all the analyzed trials occurred during an attended cueing period (as the minimum crossmodal cue duration was 2 s), and to remove transient button presses occurring early in the cue period, which were unlikely to be caused by crossmodal match or mismatch (*Figure 3—figure supplement 1*). We display the frequency range of 0–15 Hz for all conditions, as no higher frequencies (above 9 Hz) were significant after our two-stage statistical criteria.

## Statistics on spectra of first switch timing

To assess the statistical significance of behavioral spectra we used a two-stage statistical testing procedure as applied in previous investigations of attentional sampling (*Landau and Fries, 2012*) and electrophysiological research (*Maris and Oostenveld, 2007*). At the first stage, we first detected significant frequencies (at *p* < 0.005 uncorrected) through a non-parametric randomization procedure. In this procedure, after obtaining the spectral amplitude for the observed data across subjects, we generated a null distribution of first switches during the same cue period by randomly shifting switch-times within each subject, while keeping the number of perceptual switches the same. We generated 5000 surrogate datasets in this way, to test the null hypothesis that there were no temporal effects on the timing of perceptual switches. We then compared the amplitude of the Fourier transform from the observed and the surrogate data at each frequency. We regarded the spectral amplitude at a certain frequency to be significantly above chance, if the observed spectral amplitude exceeded the top 99.5% of the null-distribution of amplitudes at each frequency generated by surrogate data.

At the second stage, we applied a cluster criterion, which corrects for multiple comparisons across multiple frequencies through a permutation procedure (*Maris and Oostenveld, 2007*). We required that the first-level significance (p<0.005 uncorrected) be sustained for at least two neighboring frequencies, and retained the sum of their clustered test-statistics (amplitudes in this case) as our observed data. Then, from our surrogate dataset, we calculated the maximum cluster-based amplitudes per surrogate (maximum spectral amplitude excluding 0–1 Hz and nearest neighbor), which we retained as the null-distribution to compare against our observed data. Candidate clusters survived this second order analysis when their observed cluster-based test-statistics exceeded the top 95% of the null distribution, or corrected to $p_{cluster}$ <0.05 if testing across multiple clusters. The null-distributions obtained for our frequencies of interest in *Figure 3f* are shown in *Figure 3—figure supplement 2*.

## EEG recording and analysis

EEG was recorded at a sampling rate of 1000 Hz using three BrainAmp amplifiers and 64-channel ActiCap (BrainProducts; RRID:SCR_009443), with the impedance of each electrode kept below 10 kΩ. Ground and reference electrodes were AFz and FCz at recording, respectively. After re-referencing the data to the average of all channels, we performed linear detrending and bandpass filtering (0.1–60 Hz with a Hamming-windowed finite impulse response filter) and down-sampled the data to 250 Hz before time-frequency analysis.

We performed all time-frequency analyses using the Chronux toolbox ((http://chronux.org; *Bokil et al., 2010*); RRID:SCR_005547), and custom MATLAB scripts. To resolve our frequencies of interest (especially between 3.5 and 4.5 Hz), we used a single-taper Fourier transform with a time-window of 2 s, which resulted in a half bandwidth (*W*) of 0.5 Hz (*W* = 1/2). This half bandwidth is consequently capable of resolving differences between 3.5 and 4.5 Hz, as demonstrated in *Figures 4b* and *5b*.

## ITPC analysis

To assess the neural correlates of attentional sampling (*Figures 4* and *5*), we analyzed the inter-trial phase coherence (ITPC) within electrodes, over multiple time-frequency points (*Bastos and Schoffelen, 2015*). ITPC is an amplitude-normalized measure of the degree to which EEG responses are phase-locked to the onset of an exogenous cue, ranging between 0 (random phase over trials) and 1 (perfect phase consistency over trials). To compute ITPC, the consistency of phase angles is computed as the length of the average of unit phase vectors in the complex plane over trials. For a given time, *t*, and frequency, *f*,

$$\mathrm{ITPC(t, f)} = \left| \frac{1}{N} * \sum_{n=1}^{N} e^{i(\theta(t, f, n))} \right|$$

where *N* is the number of trials, and $\theta$ is the phase angle at time *t*, frequency *f*, and trial *n*.

Due to the stochastic nature of perceptual alternations during binocular rivalry, the number of available trials for analysis in each mismatched and matched cue type ranged from 12 to 36 trials

across subjects (after averaging first across subjects, the mean number of trials was 24 (±1.5) trials across matched/mismatched and attention conditions). Because the bias level (or expected chance level for pure noise data) of ITPC is strongly influenced by the number of trials, we took additional measures to equate the number of mismatched and matched cue types for the analysis. To achieve this aim, the minimum number of trials recorded for a given cue combination was identified across subjects. Following this, subjects with greater numbers of trials had their observed number of trials supplemented by downsampling with replacement from their recorded trials, equating them to the predefined minimum for each condition. Upon this resampled dataset, the ITPC was computed, and this process repeated 100 times. As the difference in ITPC between auditory, tactile, and combined auditory and tactile cues was not significant, we proceeded by combining crossmodal cue types within each subject.

## ITPC statistics

To investigate the neural correlates of attentional sampling, we analysed evoked ITPC, the increase in ITPC during 0 to 2 s after onset compared to −2 to 0 s before onset. Similar to our statistical approach for the behavioral spectral analysis described above, we used a two-stage statistical testing procedure for this analysis. At the first stage, we performed a *t*-test (two-tailed) comparing whether evoked 3.5 and 8 Hz ITPC differed between mismatched and matched conditions across subjects at each electrode. At each electrode, we used the mean evoked ITPC value obtained from the downsampling method described above. As a result of the *t*-tests, if we found a cluster of at least two neighboring electrodes with the same *t*-score polarity at $p < 0.05$ (uncorrected), where inter-electrode distance did not exceed 3.5 cm, we proceeded using this cluster in the second stage of statistics. As a result of this cluster criterion, we always identified a minimum size of 2-electrode clusters (*Figures 4a* and *5a*).

At the second stage, we first computed the absolute value of the sum of observed *t*-scores within the identified cluster, which we retained as our observed test-statistic (*Figure 4—figure supplement 1*; *Figure 5—figure supplement 1*). To create the null distribution, condition labels (mismatch and match) were randomly shuffled for each electrode within each subject, to create two surrogate datasets the same size as our original mismatch and match conditions. Then the *t*-scores were computed for each electrode based on our surrogate datasets, and the electrode with the maximum *t*-score and the maximum *t*-score of its neighbors retained. The sum of these *t*-scores were then retained per shuffle, and this procedure repeated 2000 times to obtain a null distribution of the sum of *t*-scores around the maximum electrode for each shuffle of our surrogate data. Against this distribution, the sum of observed *t*-scores for the candidate cluster was then compared. When the observed sum of *t*-scores was within the top 5% (or cluster corrected to $p < 0.05$) then we concluded that there was a significant difference between mismatch and match conditions. The null-distributions and observed test-statistics produced by this analysis are shown in *Figure 4—figure supplement 1* and *Figure 5—figure supplement 1*.

## Data availability

The raw data in this study are available via the Monash University Figshare repository (https://figshare.com/projects/Crossmodal_binocular_rivalry_attention_sampling_project/56252). Analysis code is available via GitHub (*Davidson, 2018*; copy archived at https://github.com/elifesciences-publications/BRproject-attentionsampling).

## Acknowledgements

MJD was supported by an Australian Government Research Training Program Scholarship. DA was funded by an Australian Research Council Discovery Project (DP150101731). NT was funded by an Australian Research Council Future Fellowship (FT120100619) and Discovery Project (DP130100194). The authors thank Bryan Paton and Claudia Lunghi for technical advice, and Brandon Lam for early piloting of the experimental paradigm.

# Additional information

## Funding

| Funder | Grant reference number | Author |
|---|---|---|
| Australian Research Council | FT120100619 | Naotsugu Tsuchiya |
| Australian Research Council | DP130100194 | Naotsugu Tsuchiya |
| Australian Research Council | DP150101731 | David Alais |

The funders had no role in study design, data collection and interpretation, or the decision to submit the work for publication.

## Author contributions

Matthew J Davidson, Conceptualization, Data curation, Formal analysis, Investigation, Visualization, Methodology, Writing—original draft, Project administration, Writing—review and editing; David Alais, Conceptualization, Resources, Software, Supervision, Methodology, Writing—review and editing; Jeroen JA van Boxtel, Naotsugu Tsuchiya, Conceptualization, Resources, Supervision, Funding acquisition, Methodology, Writing—review and editing

## Author ORCIDs

Matthew J Davidson http://orcid.org/0000-0002-2088-040X
David Alais http://orcid.org/0000-0002-0411-940X
Jeroen JA van Boxtel http://orcid.org/0000-0003-2643-0474
Naotsugu Tsuchiya http://orcid.org/0000-0003-4216-8701

## Ethics

Human subjects: This research involved human subjects. Participants gave their written informed consent to participate in the experiment. Experimental procedures were approved by the Monash University Human Research Ethics Committee (CF12/2542 - 2012001375)

## Decision letter and Author response

Decision letter https://doi.org/10.7554/eLife.40868.035
Author response https://doi.org/10.7554/eLife.40868.036

# Additional files

## Supplementary files

• Transparent reporting form
DOI: https://doi.org/10.7554/eLife.40868.023

## Data availability

The raw data in this study are available via the Monash University Figshare repository (https://figshare.com/projects/Crossmodal_binocular_rivalry_attention_sampling_project/56252). Analysis code is available via GitHub (https://github.com/Davidson-MJ/BRproject-attentionsampling; copy archived at https://github.com/elifesciences-publications/BRproject-attentionsampling).

The following dataset was generated:

| Author(s) | Year | Dataset title | Dataset URL | Database and Identifier |
|---|---|---|---|---|
| Matthew J Davidson | 2018 | Crossmodal binocular rivalry, attention sampling project | https://figshare.com/projects/Crossmodal_binocular_rivalry_attention_sampling_project/56252 | figshare, Crossmodal_binocular_rivalry_attention_sampling_project/56252 |

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
