## [Decision Letter]

Thank you for submitting your article "Attention periodically samples competing stimuli during binocular rivalry" for consideration by *eLife*. Your article has been reviewed by three peer reviewers, and the evaluation has been overseen by a Reviewing Editor and Michael Frank as the Senior Editor. The following individual involved in review of your submission has agreed to reveal his identity: Daniel H Baker (Reviewer #3).

The reviewers have discussed the reviews with one another and the Reviewing Editor has drafted this decision to help you prepare a revised submission.

Summary:

This is an innovative, interesting and timely study that reveals periodic switching dynamics during binocular rivalry, following cross-modal cues, and explores the rate of attentional sampling during dominance and suppression. Data from perceptual responses show that with relevant cross-modal cues attentional modulations are periodic. When the cue is congruent with the current visual percept, switches display an 8Hz periodicity; when the cue is incongruent, switches occur in about half of 8Hz, with a 3.5Hz frequency. This pattern of results fits well with the idea of attentional sampling, whereby attention samples a single target at around 8Hz, but for two competing targets the effective sampling rate is divided, i.e. about 4Hz. The presence of EEG correlates of these rhythmic dynamics at the same frequencies tends to support the authors' claims.

Addressing the following points will provide information regarding the validity of the claims and strengthen the manuscript.

Essential revisions:

Theoretical issues

1) You should discuss the Dieter, Melnick and Tadin, 2015 study, which shows that stimulus-driven cues can bias perceptual report. Exogenous feature-based cues were presented close to the rivalry stimuli; the cue congruent with the (currently) dominant percept lengthened its duration whereas the cue congruent with the suppressed percept hastened it to emerge back. The authors should stress the novelty of their study regarding the oscillatory nature of such process.

2) The authors should discuss whether they think that the oscillatory behavioral responses they found reflect an oscillation of perception or decision criteria, even though these two aspects cannot be distinguished in the perceptual report used in this study. Germane to this discussion is a recent study that using signal detection theory found that alpha oscillation modulates criterion, not sensitivity (Lemi et al., 2017).

3) The reported effects only occur for the 4.5-Hz modulated stimulus, and not for the 20Hz modulation. Does this imply an upper temporal limit for crossmodal interactions or for attention?

4) The authors should relate their findings to computational models of binocular rivalry. Many use mutual inhibition, adaptation and noise to describe the neural circuit underlying rivalry (e.g. Wilson, 1997; Laing and Chow, 2002). Some have explicitly implemented an attentional component (e.g. Li et al., 2017). None of the rivalry models have attempted to model oscillatory perceptual rhythms. Could the current results inform which model component(s) (e.g. input drive, mutual inhibition or attention) may be oscillatory?

5) The authors suggest that the 7-8 Hz oscillation reflects a single attentional focus, and the ~4Hz oscillation reflects attention that was split into two foci. Does that imply that without a mismatched cue, observers' attention would be predominantly allocated to the dominant image without oscillating between the two stimuli? Were that the case, how would this relate to the theory on rivalry. Are the authors suggesting that a mismatch cue allowed attention to start jumping across two items? And that such a process would not happen without a mismatched cue?

Methods, Analyses and Results

6) It is not clear that there are 3 types of cross-modal cues (auditory, tactile, and both combined) until the Materials and methods section; i.e., well after reading the Results and considering the figures. The existence of 3 cue types should be mentioned explicitly earlier. In addition, this begs the question of possible differences among the cue types: Do they affect the dynamics of switching and the results described in all figures?

7) It would be good to report the frequency tagged responses to the rivaling stimuli themselves. Are there increases in the response when a frequency-matched cross-modal cue is presented? Do the amplitudes correspond to participant button presses consistent with Brown and Norcia, 1997?

8) For computing Fourier amplitude spectrum, the authors analyzed the time window from 0.5 to 2s from the cue onset. However, Figure 3C and 3D also show oscillatory patterns in the first 0.5s time window. Include this time, narrow the window, or explain why this range was not analyzed. Note that previous studies have removed a shorter time window after the cue (e.g. Landau and Fries, 2012; Fiebelkorn et al., 2013).

9) It is unclear whether the authors performed FFT on the averaged (across subjects) time course, or on individual time courses, and then averaged the spectrum across subjects. If the former, the latter should also be reported. It is important to ensure that the effect does not merely result from the averaging procedure.

10) A technical issue regards the EEG ITPC calculation and the logic of upsampling. We agree that it is necessary to equate the number of mismatched and matched cue types for the analysis. However, the strategy of upsampling the lowest-trial-count condition up to the trial number in the highest-trial-count condition is questionable. We are not convinced that the authors' analysis confirmed that upsampling, compared to downsampling, reduced the bias introduced when equating ITPC values across subjects. Running the Matlab code below, the authors will see that the ITPC bias caused by a difference in trial count (first figure) is not reduced when upsampling the low-trial-count condition (second figure), but disappears when downsampling the high-trial-count condition (third figure). You should repeat the analysis with downsampling, ideally multiple times to avoid spurious results.

1.%% Matlab code

2. phases = 2*pi*rand(30,1000);% low trial count condition

3. otherphases = 2*pi*rand(90,1000);% high trial count condition

4. downsampled_otherphases = otherphases(1:30,:);% downsample high trial count

5. upsampled_phases = repmat(phases,3,1);% upsample low trial count

6. ITPC = abs(mean(exp(1i*phases),1));

7. otherITPC = abs(mean(exp(1i*otherphases),1));

8. downsampledITPC = abs(mean(exp(1i*downsampled_otherphases),1));

9. upsampledITPC = abs(mean(exp(1i*upsampled_phases),1));

10. figure;

11. subplot(2,1,1); hist(ITPC); title('Original phases (low N)'); set(gca,'xlim',[0 0.4],'ylim',[0 250]); xlabel('ITPC');

12. subplot(2,1,2); hist(otherITPC); title('Original phases (high N)'); set(gca,'xlim',[0 0.4],'ylim',[0 250]); xlabel('ITPC');

13. figure;

14. subplot(2,1,1); hist(upsampledITPC); title('Upsampled phases (low N)'); set(gca,'xlim',[0 0.4],'ylim',[0 250]); xlabel('ITPC');

15. subplot(2,1,2); hist(otherITPC); title('Original phases (high N)'); set(gca,'xlim',[0 0.4],'ylim',[0 250]); xlabel('ITPC');

16. figure;

17. subplot(2,1,1); hist(ITPC); title('Original phases (low N)'); set(gca,'xlim',[0 0.4],'ylim',[0 250]); xlabel('ITPC');

18. subplot(2,1,2); hist(downsampledITPC); title('Downsampled phases (high N)'); set(gca,'xlim',[0 0.4],'ylim',[0 250]); xlabel('ITPC');

19.% % End Matlab code

Figures

11) Figures 2A, 3A, 4B and 5B require figure legends so that the meaning of the colors is clear without reading the caption.

12) In Figures 4A and 5A it would be helpful to mark all electrode locations with small dots. We assume these plots are thresholded, such that electrodes with non-significant t-values are set to t=0, but this wasn't stated explicitly. Why not plot all t-values and indicate which are significant in some other way (e.g. with superimposed symbols).

13) The 9Hz difference in Figure 5B is somewhat unexpected. This is described as a 'stimulus harmonic', but if so, we wouldn't expect to see a difference between matched and mismatched trials. Do the authors have an explanation for this? Does the 9Hz response also correlate with the PSI measure?

14) The model diagrams in Figure 6 are somewhat confusing. For model (a), do the authors mean that the suppressed image emerges back because attention shifts to the mismatched cue first, and thus the mismatched cue enhances the suppressed image? For model (b), it seems surprising that the presence of the green-crossmodal cue would let the attention focus move from the dominant red image to the suppressed red image, before it moves to the dominant green image…

15) For correlation scatterplots, it would be helpful to include R and p values in the plot. Are the straight lines in these plots regression lines that predict the y-values using the x-values (i.e. minimizing the error in the y-direction)? Given that both of these are dependent variables with presumably no implied causation, we recommend using Deming regression lines instead, which minimize the absolute error between the line and each data point.

[Editors' note: further revisions were requested prior to acceptance, as described below.]

Thank you for resubmitting your work entitled "Attention periodically samples competing stimuli during binocular rivalry" for further consideration at *eLife*. Your revised article has been favorably evaluated by Michael Frank (Senior Editor), a Reviewing Editor, and three reviewers.

The three reviewers and I think that you have addressed all the issues raised satisfactorily, but one. The pending issues regards the ITPC calculations and down-sampling vs. up-sampling.

We note that there is a crucial difference in the definition of ITPC "bias": You are talking about minimizing absolute bias (the fact that ITPC is always overestimated when using a finite number of trials, to an extent inversely related to the trial number, e.g. compare Author response image 4). Instead, the reviewers are worried about relative bias (the fact that two experimental conditions with equal "true" ITPC may produce very different ITPC values if the number of trials is not equated, e.g. compare Author response image 4).

Your upsampling method keeps the absolute bias as low as possible in each condition (given the trial number), but does not reduce relative bias (e.g. compare Author response image 4). Only downsampling can remove this relative bias (e.g. compare Author response image 4). This comes with the cost of an increase in absolute bias for the high N condition (compare Author response image 4), but given that your analysis relies on a comparison of the two conditions, the relative bias must be eradicated; absolute bias changes are just collateral damage.

To summarize:

1) The reviewer made the null assumption that the 2 experimental conditions (matched/mismatched) have the same true ITPC but different numbers of trials. (Here the true ITPC was assumed to be zero, which is the worst-case scenario: no actual phase-locking).

2) When comparing ITPC across the two conditions, a systematic difference was found (Author response image 4 vs. 4D). This difference is necessarily a statistical artifact.

3) The reviewer applied the authors' upsampling method to equate trials; the systematic difference remained (Author response image 4/4C vs. 4D).

4) The reviewer applied a downsampling method to equate trials; the (artifactual) difference disappeared (Author response image 4 vs. 4E).

5) Conclusion:

The authors are correct that upsampling can preserve the ITPC to have a similar value as that derived from the original dataset (without resampling). But, this is not the point here. The aim of resampling is to let the ITPCs from the two conditions comparable when performing statistical tests. When comparing ITPC from two conditions with unequal trial numbers, upsampling can produce false positives, downsampling will not. Thus, the authors should use downsampling instead of upsampling.

---

## [Author Response]

Essential revisions:Theoretical issues1) You should discuss the Dieter, Melnick and Tadin, 2015 study, which shows that stimulus-driven cues can bias perceptual report. Exogenous feature-based cues were presented close to the rivalry stimuli; the cue congruent with the (currently) dominant percept lengthened its duration whereas the cue congruent with the suppressed percept hastened it to emerge back. The authors should stress the novelty of their study regarding the oscillatory nature of such process.

We thank the reviewers for this suggestion. We have now cited Dieter et al., 2015, in our Discussion. We have also positioned our findings in support of cue-based effects on rivalry, stressing the novel oscillatory nature of this process (Discussion, third paragraph).

2) The authors should discuss whether they think that the oscillatory behavioral responses they found reflect an oscillation of perception or decision criteria, even though these two aspects cannot be distinguished in the perceptual report used in this study. Germane to this discussion is a recent study that using signal detection theory found that alpha oscillation modulates criterion, not sensitivity (Lemi et al., 2017).

As the reviewers point out our study is unable to tell whether the oscillations in switch rates we have observed reflect changes in perceptual sensitivity or decision criteria. While the results of Iemi et al. support oscillations in decision criteria rather than sensitivity per se (Iemi et al., 2017), others show oscillations in both criterion and sensitivity (Ho et al., 2017; Zhang, Morrone, Alais, Sci Rep, In press) and perception rather than decision criteria (Iemi and Busch, 2018). We have included a short treatment of this distinction in our Discussion (ninth paragraph).

3) The reported effects only occur for the 4.5-Hz modulated stimulus, and not for the 20Hz modulation. Does this imply an upper temporal limit for crossmodal interactions or for attention?

We view it as possible that this difference reflects a temporal limit for crossmodal interactions or for attention. We have clarified our consideration of both alternatives in our Discussion (eleventh paragraph). Although high-frequency effects have been reported in other previous, similar designs (without attention), it is unclear whether our new crossmodal stimulus parameters, or the inclusion of an attentional instruction are responsible for the upper temporal limit we have observed. We believe that distinguishing between these alternatives will be a fruitful line of endeavour in future experimentation, and mention this in our Discussion also.

4) The authors should relate their findings to computational models of binocular rivalry. Many use mutual inhibition, adaptation and noise to describe the neural circuit underlying rivalry (e.g. Wilson, 1997; Laing and Chow, 2002). Some have explicitly implemented an attentional component (e.g. Li et al., 2017). None of the rivalry models have attempted to model oscillatory perceptual rhythms. Could the current results inform which model component(s) (e.g. input drive, mutual inhibition or attention) may be oscillatory?

We thank the reviewers for this suggestion, and have now incorporated a treatment of BR models in our Discussion (third paragraph). We suggest that the attention component of a rivalry model, such as the one that is modelled by Li et al., 2017, can be oscillatory. We have linked our previous discussion of van Boxtel et al., 2008; to an oscillatory model of BR more explicitly (Discussion, twelfth paragraph).

5) The authors suggest that the 7-8 Hz oscillation reflects a single attentional focus, and the ~4Hz oscillation reflects attention that was split into two foci. Does that imply that without a mismatched cue, observers' attention would be predominantly allocated to the dominant image without oscillating between the two stimuli? Were that the case, how would this relate to the theory on rivalry. Are the authors suggesting that a mismatch cue allowed attention to start jumping across two items? And that such a process would not happen without a mismatched cue?

We thank the reviewers for the opportunity to address this clarification, and would like to clarify two points: 1) We do not believe that switches in rivalry are caused only by an attentional sampling mechanism, and 2) nor that attentional switches are completely determined by an 8 / 3.5 Hz rhythm.

As to 1), there are many different types of attention (Koch and Tsuchiya, 2007), and in the context of rivalry, it is unclear the extent to which attention (or type thereof), may mediate perceptual outcome. If anything, both endogenous and exogenous effects of attention are rather limited (Alais and Blake, 2005; Maier, Panagiotaropoulos, Tsuchiya, and Keliris, 2012; Dieter et al., 2015).

As to 2), we view our results to be consistent with previous investigations of attentional sampling. Specifically, that attentional sampling rhythms modulate, but do not completely determine perceptual outcome. We have now clarified this nuance in our Discussion (sixth paragraph).

Methods, Analyses and Results6) It is not clear that there are 3 types of cross-modal cues (auditory, tactile, and both combined) until the Materials and methods section; i.e., well after reading the Results and considering the figures. The existence of 3 cue types should be mentioned explicitly earlier. In addition, this begs the question of possible differences among the cue types: Do they affect the dynamics of switching and the results described in all figures?

Thank you for this suggestion. We now point out specifically that there were three crossmodal cue types more clearly (subsection “Attending to low-frequency crossmodal stimulation promotes the perceptual dominance of low-frequency flicker during binocular rivalry”, first paragraph).

We found no difference in the crossmodal cue effects based on cue type. That is to say, whether attending to low-frequency auditory, tactile, or combined auditory and tactile cues, the results were the same. We observed an increased probability of perceiving the congruent, low-frequency flicker. A new figure shows these individual time-series, separated by crossmodal modality (Author response image 1). No significant difference between cue types emerged when comparing these time-series (repeated measures ANOVA at each time point, with FDR correction for multiple comparisons).

As the cue differences were not statistically significant, we continued by pooling them for our subsequent behavioural analyses. This had the advantage of increasing the number of trials available per mismatch x match condition, and improved our statistical power. Without pooling across cue types, the number of the trials per cue type are only ~33% of the total number of trials, and not enough to obtain significant results with our spectral analysis for first perceptual switches.

**Author response image 1. respfig1:** The effect of crossmodal cue modality on behavioural switch dynamics. Crossmodal cue modalities are shown for auditory cues in magenta, tactile cues in black and auditory plus tactile in blue. The probability of perceiving congruent flicker during attended low-frequency cues (related to manuscript Figure 2A). No significant difference between cue modalities emerged when comparing the time-course of this effect (repeated measures ANOVA at each time-point followed by FDR correction for multiple comparisons).

We also compared the ITPC values obtained at attentional sampling frequencies between our three crossmodal cue types. No significant difference emerged when comparing the spatial topography of attentional sampling frequencies.

7) It would be good to report the frequency tagged responses to the rivaling stimuli themselves. Are there increases in the response when a frequency-matched cross-modal cue is presented? Do the amplitudes correspond to participant button presses consistent with Brown and Norcia, 1997?

We are hesitant to include additional SSVEP, SSAEP, or SSSEP analyses, as the additional analyses and explanation would significantly lengthen the manuscript – and it is already very dense. We would like to reiterate that attentional sampling has been proposed to be related to the optimum phase of sampling rhythms, rather than any modulation in SSVEP power.

Having said that, the answer is ‘yes’, amplitudes correspond with subject button presses consistent with previous research on frequency-tagging during rivalry (e.g. Brown and Norcia, 1997; Zhang et al., 2011; Tononi et al., 1998). We here provide an example of the change in 4.5 Hz frequency-tagged SSVEP strength with button press. The amplitudes correspond with perceptual contents, increasing when perceiving a low-frequency flicker, and decreasing when perceiving the high-frequency flicker. We also provide for the reviewer an example of increased 4.5 Hz SNR, during these perceptual switches, when accompanied by a steady-state crossmodal stimulus (Author response image 2).

**Author response image 2. respfig2:** Changes in frequency-tagged power (in the unit of signal to noise ratio, log SNR) around the time of button presses for perceptual switches. (a) In visual-only conditions, around the time of button presses, the time course of the power of EEG at the occipital areas (electrode P08) at 4.5 Hz tracks perceptual alternations, consistent with previous investigations using frequency-tagged binocular rivalry. For each subject, we computed the mean across trials, then took the average across subjects, with the error bars reflecting the standard error across subjects. The solid line is for the mean across the trials with perceptual changes from high-frequency (20 Hz) flicker to low-frequency (4.5 Hz) flicker. The dotted line is for the opposite direction (perception changing from low to high flicker). b) Comparison between the visual-only and crossmodal conditions in perceptual-switch related EEG power, but here only focusing on the trials when percepts change from a high flicker to low flicker. In crossmodal conditions, where low-frequency (4.5Hz) auditory and/or tactile stimuli were delivered to subjects, log SNR for 4.5 Hz increased compared to the perceptual switches that happened when no crossmodal stimuli were presented. Here we only used the segment around the first perceptual switch after the onset of crossmodal cue.

8) For computing Fourier amplitude spectrum, the authors analyzed the time window from 0.5 to 2s from the cue onset. However, Figure 3C and 3D also show oscillatory patterns in the first 0.5s time window. Include this time, narrow the window, or explain why this range was not analyzed. Note that previous studies have removed a shorter time window after the cue (e.g. Landau and Fries, 2012; Fiebelkorn et al., 2013).

Regardless of the specific time windows suggested by the reviewers, our conclusions would not change.

Analyzing the spectra between 0.1 and 2 seconds after cue onset, demonstrates the same results, as shown in Author response image 3. However, we reiterate our two reasons for avoiding this early time-window (0-0.5 seconds after stimulus onset):

1) Manuscript Figure 2A shows that the attentional effects in behavior emerge only beyond 0.5 seconds after cue onset, implying that the effects before 0.5 second are not due to crossmodal attentional effects.

2) Figure 3—figure supplement 1 has been included to show that a transient increase in the rate of first switches occurs within this window (between 0-0.5 s), regardless of attentional effects (i.e., in all cue types), as well as after cue offset, implying that the effects before 0.5 second are not cue dependent. Thus, we chose to avoid this early window and focus on cue-specific effects.

**Author response image 3. respfig3:** Adjusting the window for spectral analysis, from 0.5 to 2 seconds, to 0.1 to 2 seconds, produces equivalent results, with significant attentional sampling frequencies ~3.5 and 7-8 Hz.

9) It is unclear whether the authors performed FFT on the averaged (across subjects) time course, or on individual time courses, and then averaged the spectrum across subjects. If the former, the latter should also be reported. It is important to ensure that the effect does not merely result from the averaging procedure.

We would like to state that our evidence for periodic sampling cannot result from our averaging procedure. Although the same averaging procedure was used for all conditions, we observed a differentiation of periodicity only for mismatched vs matched cues, and not unattended, high-frequency, visual-only, or post-offset time-series.

The time-course of first switches we display (manuscript Figure 3) displays the proportion of all first switches after cue onset, at each time point averaged across subjects. Our FFT is performed on this average across subjects time-course.

The pattern in individual subjects is similar, though is not present for each individual. This is because the number of switches per each of 8 conditions (match vs. mismatch x low- vs. high- frequency x attended vs. unattended) is very small per subject, and the attentional effects themselves vary across subjects – as shown in manuscript Figure 2B. Furthermore, some FFT spectra from subjects with very few perceptual switches are extremely noisy, and disrupts any effects that could be observed at the group level if averaging individual spectra.

We have now stated that this effect is clearest in the across-subject time-course, and that this analysis was chosen to match with previous investigations of attentional sampling, which also analyzed the time-course of data at the across-subject level (e.g., Landau and Fries, 2012; Fiebelkorn et al., 2013). These revisions are in the last paragraph of the subsection “Binocular rivalry dynamics during attended low-frequency crossmodal cues”.

10) A technical issue regards the EEG ITPC calculation and the logic of upsampling. We agree that it is necessary to equate the number of mismatched and matched cue types for the analysis. However, the strategy of upsampling the lowest-trial-count condition up to the trial number in the highest-trial-count condition is questionable. We are not convinced that the authors' analysis confirmed that upsampling, compared to downsampling, reduced the bias introduced when equating ITPC values across subjects. Running the Matlab code below, the authors will see that the ITPC bias caused by a difference in trial count (first figure) is not reduced when upsampling the low-trial-count condition (second figure), but disappears when downsampling the high-trial-count condition (third figure). You should repeat the analysis with downsampling, ideally multiple times to avoid spurious results.1.% % Matlab code2. phases = 2*pi*rand(30,1000);% low trial count condition3. otherphases = 2*pi*rand(90,1000);% high trial count condition4. downsampled_otherphases = otherphases(1:30,:);% downsample high trial count5. upsampled_phases = repmat(phases,3,1);% upsample low trial count6. ITPC = abs(mean(exp(1i*phases),1));7. otherITPC = abs(mean(exp(1i*otherphases),1));8. downsampledITPC = abs(mean(exp(1i*downsampled_otherphases),1));9. upsampledITPC = abs(mean(exp(1i*upsampled_phases),1));10. figure;11. subplot(2,1,1); hist(ITPC); title('Original phases (low N)'); set(gca,'xlim',[0 0.4],'ylim',[0 250]); xlabel('ITPC');12. subplot(2,1,2); hist(otherITPC); title('Original phases (high N)'); set(gca,'xlim',[0 0.4],'ylim',[0 250]); xlabel('ITPC');13. figure;14. subplot(2,1,1); hist(upsampledITPC); title('Upsampled phases (low N)'); set(gca,'xlim',[0 0.4],'ylim',[0 250]); xlabel('ITPC');15. subplot(2,1,2); hist(otherITPC); title('Original phases (high N)'); set(gca,'xlim',[0 0.4],'ylim',[0 250]); xlabel('ITPC');16. figure;17. subplot(2,1,1); hist(ITPC); title('Original phases (low N)'); set(gca,'xlim',[0 0.4],'ylim',[0 250]); xlabel('ITPC');18. subplot(2,1,2); hist(downsampledITPC); title('Downsampled phases (high N)'); set(gca,'xlim',[0 0.4],'ylim',[0 250]); xlabel('ITPC');19.% % End Matlab code

We thank the reviewers for providing Matlab code to compare upsampling and downsampling of ITPC. While the reviewers interpretation of their code appears to contradict with our original claim, in fact, it is consistent with what we have claimed: downsampling enhances the bias while upsampling does not. The discordance in interpretation comes from the way the reviewer contrasts the results (ITPC vs. otherITPC, upsampled ITPC vs. otherITPC, and ITPC vs. downsampled ITPC). As we explain in the following, we claim that the appropriate comparison is between ITPC vs upsampled ITPC and other ITPC vs. downsampled ITPC. The details in response to the reviewers’ code are as follows:

Explanation of the code:

Lines 2-5 prepare the data, in which the true ITPC is assumed as 0. We interpret *phases* to represent 30 trials sampled from 1000 channels (or subjects). Then, line 10-11 plots the expected “bias” distribution over 1000 channels where the true ITPC is 0 and sampled over 30 trials in each channel. (Reproduced as Author response image 4)

Now line 13-14 plots what the reviewers call *upsampled_phases.* But line 5 creates it by duplicating the same random distribution 3 times (Author response image 4, which is identical to Author response image 4 by definition). This is different from what we did, which is upsampling with replacement. Our procedure would replace line 5 as follows:

for iChannel = 1:1000

upsampled_phasesv2 = phases(randi(30,1, 90), iChannell);

end

With this correction, upsampling minimally affects the distribution (Author response image 4) from the original distribution (Author response image 4). This was exactly our point. Upsampling does not introduce further bias into the ITPC estimated from all available 30 trials. From the originally sampled 30 trials, we cannot (and should not be able to) reduce the bias no matter how much upsampling we do.

Next, line 12 plots the expected ITPC distribution of *otherphases* defined in line 3, where now each channel is sampled with 90 trials. This proves the point that ITPC bias (when the true ITPC is 0) reduces as we increase the number of sample trials. (Reproduced as Author response image 4)

Line 4 downsamples 30 trials out of the 90 available trials in *otherphases.* Line 18 plots this downsampled distribution of the ITPC, which proves our point that downsampling introduces further bias (comparing 4D to 4E). This is the reason why we have not used downsampling.

The above code by the reviewer establishes that when the true ITPC is assumed to be 0, 1) we should sample more trials to reduce the bias in ITPC, 2) upsampling with replacement does not introduce much bias, and 3) downsampling introduces significant bias.

Given that our data consists of N=34 subjects, whose minimal number of trials is 12 and maximal is 36, and that the mean ITPC across subjects should be computed with an equal number of trials, we opted for upsampling to the maximum number of trials across subjects.

The revised code is reproduced here for convenience:

1.% % Matlab code

2, phases = 2*pi*rand(30,1000);% low trial count condition

3, otherphases = 2*pi*rand(90,1000);% high trial count condition

4, downsampled_otherphases = otherphases(1:30,:);% downsample high trial count

5. upsampled_phases = repmat(phases,3,1);% upsample low trial count

6. upsampled_phasesv2 =zeros(size(upsampled_phases));

7, for iChannel = 1:1000

8. upsampled_phasesv2(:,iChannel) = phases(randi(30,1, 90), iChannel);

9. end

10. ITPC = abs(mean(exp(1i*phases),1));

11. otherITPC = abs(mean(exp(1i*otherphases),1));

12. downsampledITPC = abs(mean(exp(1i*downsampled_otherphases),1));

13. upsampledITPC = abs(mean(exp(1i*upsampled_phases),1));

14. upsampledITPCv2 = abs(mean(exp(1i*upsampled_phasesv2),1));

15. figure;

16. subplot(5,1,1); hist(ITPC); title('Original phases (low N)'); set(gca,'xlim',[0 0.4],'ylim',[0 250]); xlabel('ITPC');

17.% figure;

18. subplot(5,1,2); hist(upsampledITPC); title('Upsampled phases (low N)'); set(gca,'xlim',[0 0.4],'ylim',[0 250]); xlabel('ITPC');

19. subplot(5,1,3); hist(upsampledITPCv2); title('Upsampled phases v2 (low N)'); set(gca,'xlim',[0 0.4],'ylim',[0 250]); xlabel('ITPC');

20. subplot(5,1,4); hist(otherITPC); title('Original phases (high N)'); set(gca,'xlim',[0 0.4],'ylim',[0 250]); xlabel('ITPC');

21. subplot(5,1,5); hist(downsampledITPC); title('Downsampled phases (high N)'); set(gca,'xlim',[0 0.4],'ylim',[0 250]); xlabel('ITPC');

22.% % End Matlab code

**Author response image 4. respfig4:** Results from Matlab code comparing downsampling and upsampling measures on ITPC bias. The critical comparisons are between 4A and 4C, and 4D and 4E. Compared to the original ITPC values with a low trial count (4A), upsampling introduces a minimal bias (4C). Compared to original ITPC values with a high trial count (4D) downsampling introduces a large bias (4E).

As a sanity check, we performed a further analysis using our own data. Author response image 5 shows the results from an example subject, when the number of repetitions is sufficient (100 bootstraps in the example below), the statistical results using:

a) bootstrapping with replacement to retain the original number of trials (N=22 trials for this subject);

b) bootstrapping with replacement when downsampling the number of trials to N=12 (the minimum number across 34 subjects); and

c) bootstrapping with replacement when upsampling the number of trials to N=36 (the maximum across subjects),

All result in similar conclusions.

Critically however, the values for baseline ITPC (or lowest ITPC values) are highest (~0.35) with b) downsampling, compared to upsampling (~0.2) or 22 trials (~0.22), which is consistent with what we expect from the simulation code provided by the reviewer.

**Author response image 5. respfig5:** Subject level 3.5 Hz ITPC after bootstrapping with replacement for attended low-frequency auditory and tactile cues. (a) The observed topographic ITPC after bootstrapping with replacement using the observed number of trials for this subject and cue-type (n = 22, bootstrapped 100 times). b and c) Topographic ITPC b) with downsampling (12 trials) and bootstrapping with replacement and c) with upsampling (36 trials) and bootstrapping with replacement.

11) Figures 2A, 3A, 4B and 5B require figure legends so that the meaning of the colors is clear without reading the caption.

We have now added these figure legends.

12) In Figures 4A and 5A it would be helpful to mark all electrode locations with small dots. We assume these plots are thresholded, such that electrodes with non-significant t-values are set to t=0, but this wasn't stated explicitly. Why not plot all t-values and indicate which are significant in some other way (e.g. with superimposed symbols).

We had thresholded non-significant electrodes in these figures. We had elected not to superimpose electrodes onto our topoplots that would obscure significant t-scores, and similarly, to threshold all non-significant regions after our two-stage spatial cluster-based analysis. This is because isolated regions (single electrodes) were present in the topography, but did not survive our cluster-based corrections for statistical significance.

In Author response image 6 we display the non-thresholded version for the reviewers. However, we believe it is not a recommended statistical practice to show non-significant results (after corrections for multiple comparisons), thus we prefer to keep this figure only here. Having said that, we will follow the advice from *eLife* and the reviewers.

**Author response image 6. respfig6:** Alternate display of topoplots in manuscript Figures 4 and 5. At the first stage of analysis, electrodes which are significant at p <.05 uncorrected were identified (shown in grey). Those which survived our spatial cluster-based criterion to account for multiple comparisons across electrodes are shown in white.

13) The 9Hz difference in Figure 5B is somewhat unexpected. This is described as a 'stimulus harmonic', but if so, we wouldn't expect to see a difference between matched and mismatched trials. Do the authors have an explanation for this? Does the 9Hz response also correlate with the PSI measure?

We are not surprised by the 9 Hz response, as our data also shows that the 9 Hz response (frequency-double of 4.5 Hz frequency tag) tracks perceptual alternations during binocular rivalry (Author response image 7).

**Author response image 7. respfig7:** Frequency-tagged changes during binocular rivalry at PO8. SNR at 9 Hz during visual-only conditions tracks the contents of consciousness during a change to the low-frequency (4.5 Hz) percept.

During mismatched cues, predominantly, a change in percept is experienced, whereby a low frequency percept enters into conscious awareness (as shown in manuscript Figure 2C). These switches result in changes to the frequency-tagged SNR, and as a result, we see an increase in 9 Hz SNR after the onset of a mismatched cue in parieto-occipital regions.

Thus an increase in ITPC at 9 Hz is to be expected for mismatched relative to matched cues, given an increase in SNR at 9 Hz upon changes in visual percepts. Indeed, increases in SSVEP strength have been shown to result from increased ITPC (Kim et al., NatNeuro, 2007).

By contrast, during matched cues, usually no change in percept is experienced. As a result, there is no transient increase in the 9 Hz response.

Neither the 4.5 Hz response or the 9 Hz response correlates with our PSI measures.

14) The model diagrams in Figure 6 are somewhat confusing. For model (a), do the authors mean that the suppressed image emerges back because attention shifts to the mismatched cue first, and thus the mismatched cue enhances the suppressed image? For model (b), it seems surprising that the presence of the green-crossmodal cue would let the attention focus move from the dominant red image to the suppressed red image, before it moves to the dominant green image.

We included this figure to offer a schematic to emphasise the different foci of attention in our suggested interpretations, and not to provide a detailed model of how we think it works. We have clarified our intention in our revised figure legend, emphasizing that this scheme does not imply any detailed mechanisms of how attentional sampling during binocular rivalry should work. Our take-home message was to compare a) sampling between visual and crossmodal cues and b) sampling between dominant and suppressed visual stimuli. We could remove this figure if the editor and reviewers recommend.

15) For correlation scatterplots, it would be helpful to include R and p values in the plot. Are the straight lines in these plots regression lines that predict the y-values using the x-values (i.e. minimizing the error in the y-direction)? Given that both of these are dependent variables with presumably no implied causation, we recommend using Deming regression lines instead, which minimize the absolute error between the line and each data point.

We have now included r and p values in these plots.

The reviewer is correct, the straight lines are least-squares regression lines, predicting the change in PSI based on nITPC, while minimizing error in the y-direction (change in PSI). We do however have an implied causation (predicting the y-values from the x-values), as we are correlating the nITPC strength 0:2 s after stimulus onset to the change in PSI 2:4 s after cue onset.

As an alternative, we present (Author response image 8) the original correlations, with three-regression lines, as per Huang et al. (*eLife*, 2013).

Given our prior expectation of the implied causality, we would prefer to stick with the linear regression lines, and have amended our figure captions to clarify these.

**Author response image 8. respfig8:** Differentiation between Deming (orthogonal) regression and linear least-squares regression for our critical comparisons, related to manuscript Figures 4 and 5. In all panels, Deming regression lines are plotted in black. Regression lines for horizontal residuals are displayed in green.

[Editors' note: further revisions were requested prior to acceptance, as described below.]

The three reviewers and I think that you have addressed all the issues raised satisfactorily, but one. The pending issues regards the ITPC calculations and down-sampling vs. up-sampling.We note that there is a crucial difference in the definition of ITPC "bias": You are talking about minimizing absolute bias (the fact that ITPC is always overestimated when using a finite number of trials, to an extent inversely related to the trial number, e.g. compare Author response image 4). Instead, the reviewers are worried about relative bias (the fact that two experimental conditions with equal "true" ITPC may produce very different ITPC values if the number of trials is not equated, e.g. compare Author response image 4).Your upsampling method keeps the absolute bias as low as possible in each condition (given the trial number), but does not reduce relative bias (e.g. compare Author response image 4). Only downsampling can remove this relative bias (e.g. compare Author response image 4). This comes with the cost of an increase in absolute bias for the high N condition (compare Author response image 4), but given that your analysis relies on a comparison of the two conditions, the relative bias must be eradicated; absolute bias changes are just collateral damage.To summarize:1) The reviewer made the null assumption that the 2 experimental conditions (matched/mismatched) have the same true ITPC but different numbers of trials. (Here the true ITPC was assumed to be zero, which is the worst-case scenario: no actual phase-locking).2) When comparing ITPC across the two conditions, a systematic difference was found (Author response image 4 vs. 4D). This difference is necessarily a statistical artifact.3) The reviewer applied the authors' upsampling method to equate trials; the systematic difference remained (Author response image 4/4C vs. 4D).4) The reviewer applied a downsampling method to equate trials; the (artifactual) difference disappeared (Author response image 4 vs. 4E).5) Conclusion:The authors are correct that upsampling can preserve the ITPC to have a similar value as that derived from the original dataset (without resampling). But, this is not the point here. The aim of resampling is to let the ITPCs from the two conditions comparable when performing statistical tests. When comparing ITPC from two conditions with unequal trial numbers, upsampling can produce false positives, downsampling will not. Thus, the authors should use downsampling instead of upsampling.

We are pleased to hear that our ITPC resampling procedure is the only remaining issue.

After reviewing your explanation, we are in agreement that reducing the relative bias in ITPC values is preferable in this case.

We have revised our analysis and proceeded by downsampling. Our overall conclusions have not changed, and we have adjusted the statistical values and figures accordingly.

These changes are in the revised manuscript, and include new data for Figures 4 and 5.

All associated source data and figure supplements have also been updated.

We have now addressed all of the reviewers’ concerns and hope our paper is now suitable for publication in *eLife.*